# ORBIT-EQUIVARIANT GRAPH NEURAL NETWORKS

**Matthew Morris, Bernardo Cuenca Grau, & Ian Horrocks**
Department of Computer Science
University of Oxford
`{matthew.morris,bernardo.grau,ian.horrocks}@cs.ox.ac.uk`

## ABSTRACT

Equivariance is an important structural property that is captured by architectures such as graph neural networks (GNNs). However, equivariant graph functions cannot produce different outputs for similar nodes, which may be undesirable when the function is trying to optimize some global graph property. In this paper, we define orbit-equivariance, a relaxation of equivariance which allows for such functions whilst retaining important structural inductive biases. We situate the property in the hierarchy of graph functions, define a taxonomy of orbit-equivariant functions, and provide four different ways to achieve non-equivariant GNNs. For each, we analyze their expressivity with respect to orbit-equivariance and evaluate them on two novel datasets, one of which stems from a real-world use-case of designing optimal bioisosteres.

## 1 INTRODUCTION

Graph neural networks (Gilmer et al., 2017; Gori et al., 2005; Wu et al., 2020) are a class of neural models designed for learning functions on graphs which have been successfully applied across a wide range of domains, including chemistry (Li et al., 2018), computer vision (Wang et al., 2019), traffic (Yao et al., 2018), recommender systems (Ying et al., 2018), combinatorial optimization (Bengio et al., 2021), and NLP (Marcheggiani et al., 2018). GNNs leverage local message-passing operations that are invariant to permutations so that the resulting output does not fundamentally depend on how the input graph is reordered. This strong inductive bias enables GNNs to effectively and efficiently learn functions combining local and global graph labels (Battaglia et al., 2018; Bronstein et al., 2021).

It is well known that common GNNs are equivalent to the Weisfeiler-Leman (1-WL) graph coloring algorithm and are thus limited in the classes of non-isomorphic graphs they can distinguish (Morris et al., 2019; Xu et al., 2019). Different models have been proposed to address these limitations, including appending unique IDs (Dasoulas et al., 2020; Loukas, 2020; Morris et al., 2022) or random noise (Abboud et al., 2021; Sato et al., 2021) to node labels, random dropouts (Papp et al., 2021), and augmenting node features or message passing using graph-distance features (Li et al., 2020). Srinivasan & Ribeiro (2019) relate structural representations obtained by GNNs and node embeddings. Zhang et al. (2021) introduce the 'labeling trick' framework to learn maximally expressive representations for sets of nodes (e.g. for link prediction). Although significant advancements have been achieved, all these models are still designed with the primary objective of enhancing the expressiveness of GNNs in the context of permutation-equivariant or permutation-invariant functions.

As pointed out by Morris et al. (2022), even GNNs with universal expressivity for equivariant functions cannot solve certain types of *symmetry breaking* graph problems. Equivariant functions cannot yield different outputs for *similar* nodes (i.e., structurally identical nodes with the same label). This presents problems in multi-agent scenarios (Morris et al., 2022) where similar agents in a communication graph might need to take different actions to solve a task (e.g., drones that need to split up to find a target). Similarly, if GNNs are used to transform molecules so as to optimize a molecular property, equivariant functions cannot transform similar atoms in different ways. An example of such a property is lipophilicity: the ability of a chemical compound to dissolve in fats, oils, and similar solvents. This is an important factor in drug design (Arnott & Planey, 2012; Broccatelli et al., 2018; Waring, 2010), as high lipophilicity often leads to undesirable effects (Hopkins et al., 2014; Obach et al., 2008). Figure 1 illustrates how optimizing the lipophilicity of a fluroxene molecule requires two out of three similar fluorine atoms to be replaced with chlorine and bromine atoms.

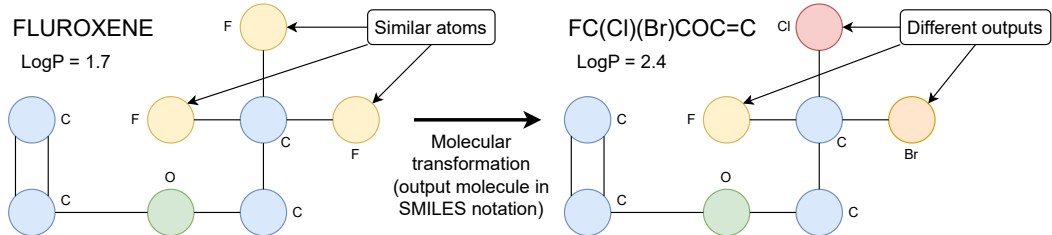

Figure 1: Non-equivariant molecular transformation that increases lipophilicity (LogP), where the nodes are labelled with the atom type, without positional information.

To tackle symmetry-breaking problems, we advocate the adoption of functions that generate a multiset of output labels for each equivalence class, called an *orbit*, formed by the node similarity relation; for instance, in Figure 1, the orbit of three fluorine atoms is replaced by a multiset of three new atoms. The assignment of labels to specific nodes should be considered arbitrary because all nodes are fundamentally identical within the same orbit. Thus, a more permissive property than equivariance is necessary, which we define as *orbit-equivariance*. This property stipulates that irrespective of how the input graph is reordered, the multiset of outputs for each distinct orbit should remain identical. This approach enables us to address symmetry-breaking problems while preserving the valuable inductive bias that "the manner in which the graph is rearranged should not impact the outcome".

Our contributions in this paper are as follows. We define orbit-equivariance and place it in the hierarchy of node-labelling functions as strictly in-between equivariant and universal node-labelling functions. We introduce a property called `max-orbit`, which allows for further refinement of orbit-equivariant functions. We establish a connection between the 1-WL graph isomorphism test and the identification of orbits, using it to assess the capacity of GNNs to differentiate between the orbits within a graph. We exploit these insights to examine four distinct GNN architectures, two of them novel, which are capable of learning non-equivariant functions and examine the theoretical expressivity of each architecture. Additionally, we define a custom loss function designed to facilitate the training of orbit-equivariant GNNs. To evaluate the performance of our models, we conduct empirical assessments on two novel datasets: Bioisostere and Alchemy-Max-Orbit. The former stems from a real-world scenario involving the optimization of drug effectiveness. Neither of these tasks can be fully addressed using equivariant functions, but both can be tackled with the aid of an orbit-equivariant function. Full proofs for all our technical results are given in Appendix A.

## 2 BACKGROUND

**Graphs**    A labelled graph $G$ is a triple $(V, E, \lambda)$, where $V(G)$ is a finite set of nodes, $E(G) \subseteq \{(v, w) \mid v, w \in V(G)\}$ is a set of directed edges, and $\lambda : V(G) \to \mathbb{L}$ is an assignment of labels from the codomain $\mathbb{L}$ to each node. For $v \in V(G)$, $\lambda(v)$ is the label (also known as an attribute, or feature) of $v$. For ease of notation, we let $V(G) = \{1, 2, ..., n\}$ for $G$ with $n$ nodes and denote $\lambda(v)$ as $G_v$. A graph is undirected if $E(G)$ is symmetric.[1] A permutation on a set $X$ is a bijection $\sigma : X \to X$. A permutation $\sigma$ on a tuple $(x_1, ..., x_n)$ yields the tuple $(\sigma(x_1), ..., \sigma(x_n))$. A permutation $\sigma$ on a graph $G$ yields the graph $H := \sigma \cdot G$, where $V(H) = V(G)$, $E(H) = \{(\sigma(v), \sigma(w)) \mid (v, w) \in E(G)\}$, and $G_v = H_{\sigma(v)}$ for each $v \in V(G)$. An automorphism of $G$ is a permutation $\sigma$ on $G$ satisfying $\sigma \cdot G = G$. Nodes $v, w \in V(G)$ are *similar* if there is an automorphism $\sigma$ of $G$ such that $\sigma(v) = w$. Node similarity induces an equivalence relation on $V(G)$, where each equivalence class is an *orbit*. The set $R(G)$ of orbits of $G$ forms a partition of $V(G)$. Figure 2 depicts an example of graph orbits.

**Equivariance**    Permutation equivariance is an important property for models realising functions on graphs (Battaglia et al., 2018; Bronstein et al., 2021). Intuitively, if the function has an output for each node, then permuting the nodes of the input should be equivalent to applying the same permutation to the output values (see Figure 3). A node-labelling function $f$ maps each $G = (V, E, \lambda)$ in its domain with $V(G) = \{1, 2, ..., n\}$ to an assignment $f(G) \in \mathbb{L}^n$ of the nodes of $G$ to values from the

---

[1]Although in this paper we focus on node labels only, we anticipate that our definitions and results will extend seamlessly to graphs with edge and global labels.

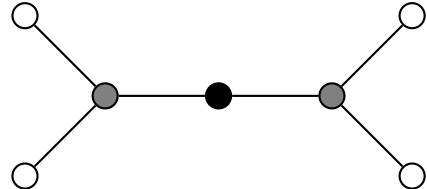

Figure 2: Graph with nodes colored by orbit.

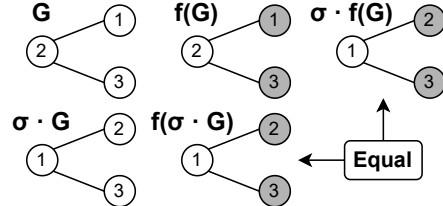

Figure 3: Example of an equivariant function $f$.

codomain $\mathbb{L}$. Node-labelling function $f$ on a domain $D$ closed under permutation is *equivariant* if $f(\sigma \cdot G) = \sigma \cdot f(G)$ holds for all $G \in D$ and permutations $\sigma$ on $V(G)$.

**Graph Neural Networks**   GNNs such as GCNs (Kipf & Welling, 2016), GraphSAGE (Hamilton et al., 2017), and GATs (Veličković et al., 2018) are commonly defined within the Message Passing framework (Gilmer et al., 2017). A GNN consists of multiple local message-passing layers that can include global readouts (Barceló et al., 2020). For layer $m$ and node $v$ with label $\boldsymbol{\lambda}_v^m$, the new label $\boldsymbol{\lambda}_v^{m+1}$ is computed as $\boldsymbol{\lambda}_v^{m+1} := f_{\text{update}}^{\theta_m}(\boldsymbol{\lambda}_v^m, \ f_{\text{aggr}}^{\theta'_m}(\{\!\!\{\boldsymbol{\lambda}_w^m \mid w \in N(v)\}\!\!\}), \ f_{\text{read}}^{\theta''_m}(\{\!\!\{\boldsymbol{\lambda}_w^m \mid w \in V(G)\}\!\!\}))$, where $N(v)$ is all nodes with edges connecting to $v$ and $\theta_m, \theta'_m, \theta''_m$ are the sets of parameters of the update, aggregation, and readout functions for layer $m$. Parameters may be shared between layers, e.g. $\theta_0 = \theta_1$. The functions $f_{\text{aggr}}^{\theta'_m}, f_{\text{read}}^{\theta''_m}$ are permutation invariant. This definition can be extended to incorporate graphs with global and edge labels (Battaglia et al., 2018). Importantly, GNNs are equivariant node-labelling functions for graphs labelled with real vectors (Hamilton, 2020).

## 3   ORBIT-EQUIVARIANCE

As discussed in Section 1, equivariance may be too restrictive in applications where optimizing some graph-level property (such as lipophilicity) requires similar nodes to be differentiated. Indeed, equivariant functions will always produce the same output for nodes within the same graph orbit.

**Proposition 1.** *Let $f$ be an equivariant node-labelling function and let $G$ be a labelled graph in its domain. If $v, w \in V(G)$ are similar, then $f(G)_v = f(G)_w$.*

We next introduce *orbit-equivariance* as a suitable relaxation of equivariance that still retains its critical structural inductive biases (Battaglia et al., 2018; Bronstein et al., 2021) while extending the range of problems that can solved. This is accomplished by mandating that, regardless of the permutation applied to the input graph, the multiset of outputs associated with each distinct graph orbit should not change.

**Definition 1.** *A node-labelling function $f$ on domain $D$ closed under permutation is **orbit-equivariant** if, for all labelled graphs $G \in D$, permutations $\sigma$ on $V(G)$, and orbits $r \in R(G)$, it holds that $\{\!\!\{f(\sigma \cdot G)_{\sigma(v)} \mid v \in r\}\!\!\} = \{\!\!\{f(G)_v \mid v \in r\}\!\!\}$.*

We now position orbit-equivariance within the hierarchy of node-labelling functions. We begin by demonstrating that not all node-labelling functions are orbit-equivariant. In the following example, we define graphs with the vertex set $V(G) = \{1, 2, 3\}$. Node colors represent their Boolean labels, where *white* represents 0 and *gray* signifies 1. While visually represented as a graph, the output of each function corresponds to a 3-tuple of node labels. Consider a node-labelling function $f$ that exhibits the following behavior for distinct graphs $G_1, G_2, G_3$ and assigns all other graphs $G$ to $0^{|G|}$.

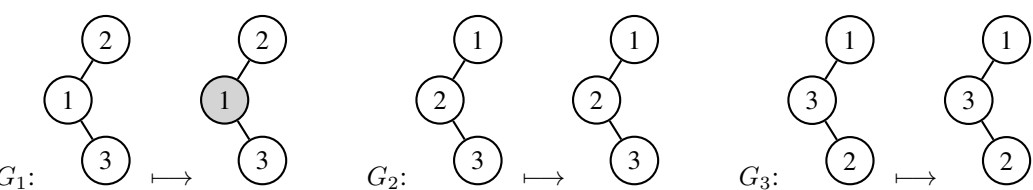

This function is not orbit-equivariant. Consider permutation $\sigma = \{(1,2),(2,1),(3,3)\}$, for which $\sigma \cdot G_1 = G_2$, and consider orbit $r = \{1\}$ of $G_1$. Then $\{\!\{f(\sigma \cdot G_1)_{\sigma(v)} \mid v \in r\}\!\} = \{\!\{f(G_2)_2\}\!\} = \{\!\{0\}\!\}$ and $\{\!\{f(G_1)_v \mid v \in r\}\!\} = \{\!\{f(G_1)_1\}\!\} = \{\!\{1\}\!\}$. So there exists an orbit $r \in R(G_1)$ and permutation $\sigma$ such that $\{\!\{f(\sigma \cdot G_1)_{\sigma(v)} \mid v \in r\}\!\} \neq \{\!\{f(G_1)_v \mid v \in r\}\!\}$.

For an example of an orbit-equivariant function that is not equivariant, consider the node-labelling function $f$ that maps input graphs $G_1, G_2, G_3$ as shown, and maps each other graph $G$ to $\{0\}^{|G|}$.

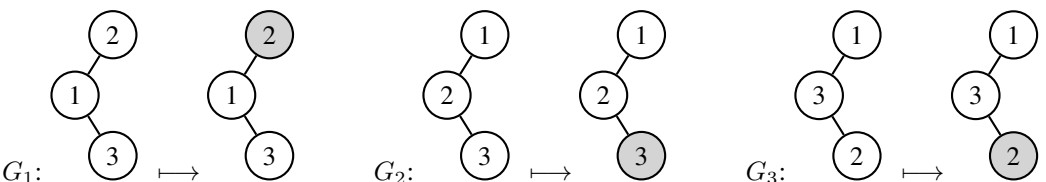

We first show that orbit-equivariance holds for $G_1$. There are two orbits of $G_1$, $r_1 := \{1\}, r_2 = \{2,3\}$. Consider the permutation $\sigma = \{(1,2),(2,1),(3,3)\}$. Note that $\sigma \cdot G_1 = G_2$ and $\sigma \cdot G_2 = G_1$. For $r_1$, we have $\{\!\{f(\sigma \cdot G_1)_{\sigma(v)} \mid v \in r_1\}\!\} = \{\!\{f(G_1)_v \mid v \in r_1\}\!\} = \{\!\{0\}\!\}$, and for $r_2$, we have $\{\!\{f(\sigma \cdot G_1)_{\sigma(v)} \mid v \in r_2\}\!\} = \{\!\{f(G_1)_v \mid v \in r_2\}\!\} = \{\!\{0,1\}\!\}$. A similar argument holds for the other graphs and permutations. Thus, $f$ is orbit-equivariant. However, it is not equivariant. For the same permutation $\sigma$, $f(\sigma \cdot G_1) = f(G_2) = (0,0,1)$ and $\sigma \cdot f(G_1) = \sigma \cdot (0,1,0) = (1,0,0)$. So $f(\sigma \cdot G_1) \neq \sigma \cdot f(G_1)$. Furthermore we can show that each equivariant function is orbit-equivariant.

**Proposition 2.** *All equivariant functions are orbit-equivariant, but not vice-versa. There exist node-labelling functions which are not orbit-equivariant.*

We proceed to establish a taxonomy of orbit-equivariant functions, categorizing them according to the count of distinct values in the output for each orbit. This refined definition offers a more granular characterization of orbit-equivariant functions, facilitating the identification of functions that might be empirically challenging to learn and aiding in the development of tools, particularly GNNs, for approximating such functions.

**Definition 2.** *For a node-labelling function $f$ on domain $D$, we define* max-orbit$(f)$ *to be the maximum across all $G \in D$ and orbits $r \in R(G)$ of the cardinality of the set $\{f(G)_v \mid v \in r\}$.*

For any equivariant function $f$, max-orbit$(f) = 1$ as a direct consequence of Proposition 1. For any $m \in \mathbb{Z}^+$, there is an orbit-equivariant function $f$ such that max-orbit$(f) = m$, since there are graphs (e.g. cycle graphs) with arbitrarily large orbits.

**Proposition 3.** *If $f$ is orbit-equivariant and* max-orbit$(f) = 1$*, then $f$ is equivariant.*

The proposition above illustrates that the only restriction lost when moving from equivariant to orbit-equivariant functions is the limitation of having a single value in the output of each orbit, which aligns with the intuition motivating orbit-equivariance. The example below of a function $f$ shows why max-orbit$(f) = 1$ alone is not sufficient to obtain equivariance. Function $f$ maps input graphs $G_1, G_2, G_3$ as shown, and maps each other graph $G$ to $\{0\}^{|G|}$. Certainly max-orbit$(f) = 1$, but the function is not equivariant, since for permutation $\sigma = \{(1,2),(2,1),(3,3)\}$, $f(\sigma \cdot G_1) = f(G_2) = (0,0,0)$ and $\sigma \cdot f(G_1) = \sigma \cdot (0,1,1) = (1,0,1)$, so $f(\sigma \cdot G_1) \neq \sigma \cdot f(G_1)$.

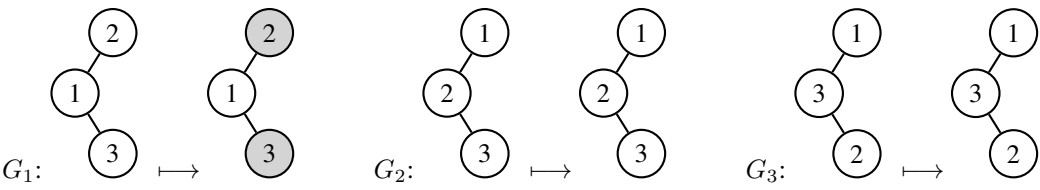

## 4 NON-EQUIVARIANT GNN ARCHITECTURES

In this section, we discuss four GNN architectures for solving orbit-equivariant problems which extend standard equivariant GNNs in a way that is independent from the specific details of the GNN.

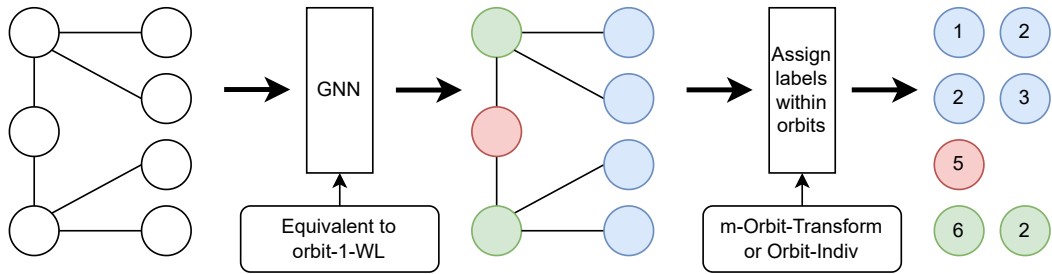

Figure 4: Architecture for constructing orbit-equivariant GNNs.

For each of these extensions, we provide a classification of their expressivity and in Appendix C.1 we analyze their computational complexity.

Before delving into the details of these architectures, we first argue that the 1-WL graph coloring algorithm (Morris et al., 2021; Weisfeiler & Leman, 1968) for isomorphism testing can also be used to compute graph orbits. This is achieved by computing iterative recolorings in the usual way, and then using the final stable colorings to partition the nodes into their orbits (instead of comparing them to another histogram of colors as in the standard isomorphism testing use case). When 1-WL is used in this manner, we refer to it as *orbit-1-WL*. Theorems 1 and 2 of Morris et al. (2019) prove that GNNs and 1-WL respectively model each other's computation, which implies that GNNs can distinguish between orbits if and only if orbit-1-WL can. That is: for nodes $v, w \in V(G)$, there exists a GNN $f$ such that $f(v) \neq f(w)$ if and only if the final colors of $v$ and $w$ are different when orbit-1-WL is applied to $G$. Orbit-1-WL is sound but not complete: there exists a graph $G$ and nodes $v, w \in G$ such that orbit-1-WL assigns the same colorings to $v$ and $w$, but $v$ and $w$ are not similar. Such graphs are shown in Appendix B.3.

**Theorem 1.** *If orbit-1-WL assigns different colorings to $u, v \in V(G)$, then $u, v$ are not similar.*

Analogously to 1-WL (Morris et al., 2019), there is a strict hierarchy of orbit-k-WL algorithms such that for all $k \geq 3$ there are graphs for which orbit-k-WL correctly identifies the orbits but orbit-(k-1)-WL does not. This is proved in Appendix B.4.

We are now ready discuss and analyze four distinct GNN architectures that are able to capture non-equivariant functions. The first two have been proposed in prior work.

(1) **Unique-ID-GNN** (Dasoulas et al., 2020; Loukas, 2020; Morris et al., 2022): this extension deterministically appends a unique ID to the label of each node before the GNN is applied to the input graph.

(2) **RNI-GNN** (Abboud et al., 2021; Sato et al., 2021): some independently sampled random noise is appended to the label of each node before the GNN is applied to the input graph; max pooling is used for the global readout $f_{\text{read}}$ in each GNN layer.

The remaining two architectures are novel and can be viewed as instantiations of the general approach illustrated in Figure 4, where we first apply a standard GNN to the input graph, then compute the orbits using orbit-1-WL, and finally devise a mechanism for assigning labels to nodes within the identified orbits.

(3) **Orbit-Indiv-GNN**: in this extension, the GNN is first applied to the input graph $G$ as usual to yield $f_{\text{GNN}}(G)$. Orbit-1-WL is subsequently used to identify the orbits of $G$; then, for each orbit, a unique ID from $\{1, ..., m\}$ is appended to each of the node labels, where $m$ is the orbit size, yielding an updated labelled graph $f_{\text{orbit\_indiv}}(f_{\text{GNN}}(G))$. A single MLP is finally applied to every extended node label in parallel and hence the overall function is defined as $f_{\text{mlp}}(f_{\text{orbit\_indiv}}(f_{\text{GNN}}(G)))$.

(4) $m$-**Orbit-Transform-GNN**: this is a more advanced extension, and we illustrate its formal description with an example depicted in Figure 5; in our description, we use (*number) to refer to markers in the example. A GNN (*1) is first applied to the input $G$ to yield $f_{\text{GNN}}(G)$ and the orbits are computed using orbit-1-WL; in the example, we demonstrate the model primarily using the orbit highlighted in blue. Each vector $f_{\text{GNN}}(G)_v$ has $m \cdot (o + 1)$ components, where $o$ is the number of final output channels (often categories) and $m \geq 2$ is a parameter of the model; as a result, we can

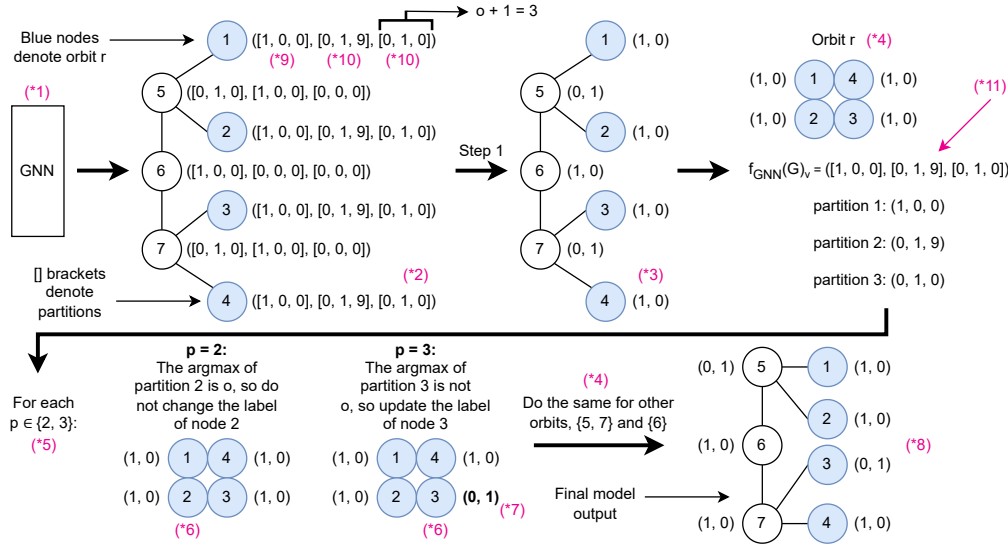

Figure 5: The transform operation of a 3-Orbit-Transform-GNN with $o = 2$.

define $m$ partitions $f_{\text{GNN}}(G)_v^1, ..., f_{\text{GNN}}(G)_v^m$ in the output of $f_{\text{GNN}}(G)_v$, each of them of size $(o+1)$ (*2). Then we apply $f_{\text{transform}}$ to $f_{\text{GNN}}(G)$, as follows.

1. Iterate through each $v \in V(G)$, and set $G_v := f_{\text{GNN}}(G)_v^1[0 : o - 1]$, where $[0 : o - 1]$ denotes the vector slice from 0 to $o - 1$ (inclusive). This a vector of size $o$ (*3).

2. Iterate through each orbit $r \in R(G)$ (*4) and each partition $p \in \{2, 3, ..., \min(m, |r|)\}$ (*5). Select the $p$-th node $v$ (*6) in orbit $r$, where $r$ is sorted by ascending order of node index. Then, if $\text{argmax}(f_{\text{GNN}}(G)_v^p) \neq o$, set $G_v := f_{\text{GNN}}(G)_v^p[0 : o - 1]$ (*7).

This yields a final model output of $f_{\text{transform}}(f_{\text{GNN}}(G))$ (*8). Since $f_{\text{GNN}}$ is equivariant, for each orbit $r \in R(G)$ and for all nodes $v, w \in r$, $f_{\text{GNN}}(G)_v = f_{\text{GNN}}(G)_w$. Thus, the transformed output for an entire orbit $r$ is essentially contained in the output of $f_{\text{GNN}}(G)_v$ for any $v \in r$. Intuitively, the first partition $f_{\text{GNN}}(G)_v^1$ (*9) contains the default value for the orbit and each remaining partition value $f_{\text{GNN}}(G)_v^p$ (*10) represents that the label of a single node $v$ in the orbit $r$ should be swapped from the default to the given value. If the final channel of the partition (with index $o$) is the largest (*11), this represents that the default should not be swapped out for the value in this partition. The motivation for this is that it aids the model in learning functions which are "nearly equivariant": i.e. ones in which within most orbits, most of the target labels within each orbit are the same.

**Theorem 2.** *The expressivity of our proposed models is as follows, where for a class of models $X$, the claim "$X$ is not equivariant" means that "there exists $f \in X$ such that $f$ is not equivariant":*

1. *Unique-ID-GNNs are not orbit-equivariant and, for any fixed $n$, can approximate any node-labelling function $f : G_n \to \mathbb{R}^n$, where $G_n$ is the set of graphs with $\leq n$ nodes.*

2. *RNI-GNNs are equivariant in expectation and, for any fixed $n$, can approximate any equivariant function $f : G_n \to \mathbb{R}^n$ with probability arbitrarily close to 1 (Abboud et al., 2021; Morris et al., 2022). They can approximate some non-equivariant and orbit-equivariant functions with probability arbitrarily close to 1, but there exist RNI-GNNs which, with probability arbitrarily close to 1, are not orbit-equivariant.*

3. *Orbit-Indiv-GNNs are not equivariant but are orbit-equivariant on graphs whose orbits are distinguishable by orbit-1-WL. For any $m \in \mathbb{Z}^+$, there exist Orbit-Indiv-GNNs $f$ with* `max-orbit`$(f) > m$.

4. *$m$-Orbit-Transform-GNNs $f$ are not equivariant but are orbit-equivariant on graphs whose orbits are distinguishable by orbit-1-WL. They have* `max-orbit`$(f) \leq m$ *and there exist $m$-Orbit-Transform-GNNs $f$ with* `max-orbit`$(f) = m$.

## 5 EXPERIMENTS

**Loss** We focus on categorical outputs in our experiments, so we adopt cross-entropy loss as our baseline. For the loss to be representative of the model performance for orbit-equivariant problems, however, we need to account for the fact that we now support a multiset of outputs for each orbit, and that treating multisets as vectors may lead to sub-optimal performance. Welleck et al. (2018) investigate loss functions for multiset prediction, but purely in the context of the multiset being a collection of sequential decisions made by a policy. Similarly to the method of Probst (2018), we use a deterministic algorithm to greedily pair up outputs and targets and then compute cross-entropy between them. Specifically, for each orbit, we sort the model outputs and target outputs by fixing some a-priori order upon the output categories, and then pair outputs and targets based on where they appear in the sorted lists. Computationally, this is $O(n \log n)$, so computing the loss is feasible even for large orbits. We refer to this method as *orbit-sorting cross-entropy loss*. For the $m$-Orbit-Transform-GNN models, loss is computed before the transform takes place, since the transform operation is not parameterised and does not require training. To obtain targets for the loss, we invert the transform operation on the ground truth outputs for the training set. This means that standard cross-entropy can be used, since the model is equivariant before the transform is applied. Full details on our novel loss function are given in Appendix C.4. We note that there are many other ways ((Zhu et al., 2016), for example) in which this problem could be approached, and leave it as future work. We focus on evaluating different models in our experiments, rather than different loss functions.

**Datasets** We propose a new dataset (**Bioisostere**) capturing a real-world use-case and a new suite of synthetic datasets (**Alchemy-Max-Orbit**) designed to test the ability of models to learn functions with different max-orbit values. Although synthetic, Alchemy-Max-Orbit datasets consist of graphs obtained from real-world organic molecule data.

The task in **Bioisostere** is, given an input molecule as a graph, output a label for each node which represents either keeping the atom or swapping it out for some other particular atom, in such a way that the final resulting molecule has minimal lipophilicity; as discussed in Section 1, this is an important factor in drug design. Solving this task requires a non-equivariant function when an atom being swapped out comes from an orbit of size $\geq 2$. To construct this dataset, we retrieved 961 small drug molecules from ChEMBL (Mendez et al., 2019) and then used MMPDB (RDKit, 2023a) to compute many different bioisosteres for each molecule that each differ by at most one atom from the original molecule. Using RDKit (RDKit, 2023b), we computed the lipophilicity of each bioisostere and selected the one that yielded the lowest value as the target for each molecule. Of these molecules, only 588 actually require some swapping out of an atom to achieve minimal lipophilicity, and only 156 require a non-equivariant function. The function $f$ required to solve the task is orbit-equivariant with `max-orbit`$(f) = 2$.

An **Alchemy-Max-Orbit-**$m$ dataset is constructed using Alchemy (Chen et al., 2019; Morris et al., 2020), a large dataset of organic molecules. All graphs without an orbit of size $\geq m$ are removed and the dataset is then extended by augmenting the existing graphs, if it is not already large enough. Details on this process are in Appendix C.2. The task in the dataset is to identify the largest orbit(s) in the graph and, for each largest orbit, distribute the labels $T := \{\!\{1, 1, ..., 1, 1, 2, ..., m\}\!\}$ at random without replacement amongst the orbit's nodes, such that $|T|$ coincides with the number of nodes in the orbit. Every node not part of a largest orbit should be assigned the label of $0$. Equivariant functions such as GNNs cannot solve any of the examples in the dataset. The function $f$ required to solve the task is orbit-equivariant with `max-orbit`$(f) = m$.

**Methodology** We perform our experiments on the Bioisostere, Alchemy-Max-Orbit-2, and Alchemy-Max-Orbit-6 datasets. We adopt GCNs (Kipf & Welling, 2016) as our baseline GNN to be augmented, since they are simple, effective, and still competitive in contemporary work (Frasca et al., 2022; Hou et al., 2022; Li et al., 2022; Rampášek et al., 2022). For each experiment, we train and test the following models: Unique-ID-GCN, RNI-GCN, Orbit-Indiv-GCN, and $m$-Orbit-Transform-GCN. Pure GCNs are equivariant and thus cannot achieve a better graph accuracy than 0 on the Alchemy-Max-Orbit datasets. However, they can still solve some subset of the examples in Bioisostere, so we include it as a baseline to see how an equivariant model performs. Every experiment is run 10 times, with different seeds, and 10% of the data is randomly left out of training to be used as a test dataset. We report the mean and standard deviation across all seeds. We run

Table 1: Mean and standard deviation of final model accuracy percentage on the test datasets.

| Dataset | Model | Graph Accuracy | Orbit Accuracy | Node Accuracy |
|---|---|---|---|---|
| **Bioisostere** (cross-entropy loss) | GCN | $52.4 \pm 6.37$ | $92.9 \pm 1.14$ | $94.4 \pm 0.79$ |
| | Unique-ID-GCN | $66.1 \pm 5.13$ | $94.5 \pm 0.97$ | $95.6 \pm 0.62$ |
| | RNI-GCN | $63.6 \pm 4.29$ | $93.9 \pm 0.86$ | $95.1 \pm 0.76$ |
| | Orbit-Indiv-GCN | $\mathbf{69.9 \pm 4.68}$ | $\mathbf{95.4 \pm 0.63}$ | $\mathbf{96.3 \pm 0.49}$ |
| | 2-Orbit-Transform | $57.1 \pm 6.43$ | $93 \pm 0.99$ | $94.1 \pm 0.79$ |
| **Alchemy-Max-Orbit-2** (orbit-sorting cross-entropy) | Unique-ID-GCN | $20 \pm 4.57$ | $79.9 \pm 2.01$ | $77 \pm 1.52$ |
| | RNI-GCN | $0 \pm 0$ | $74.5 \pm 1.7$ | $75.2 \pm 1.66$ |
| | Orbit-Indiv-GCN | $\mathbf{51.9 \pm 4.38}$ | $\mathbf{87.5 \pm 1.78}$ | $\mathbf{90.6 \pm 1.12}$ |
| | 2-Orbit-Transform | $47.9 \pm 6.45$ | $86.8 \pm 1.53$ | $85.1 \pm 1.66$ |
| **Alchemy-Max-Orbit-6** (orbit-sorting cross-entropy) | Unique-ID-GCN | $66.8 \pm 7.15$ | $84.8 \pm 2.97$ | $95.4 \pm 1.07$ |
| | RNI-GCN | $44.9 \pm 7.19$ | $78.5 \pm 3.39$ | $91.4 \pm 1.47$ |
| | Orbit-Indiv-GCN | $\mathbf{83.4 \pm 4.22}$ | $\mathbf{88.9 \pm 2.71}$ | $\mathbf{97.1 \pm 1.46}$ |
| | 6-Orbit-Transform | $10.6 \pm 4.14$ | $71.2 \pm 2.47$ | $87.6 \pm 1.08$ |

each experiment with standard cross-entropy loss and with orbit-sorting cross-entropy loss. Full hyperparameters for our experiments can be found in Appendix C.3. Accuracy is measured and reported in 3 ways: the average proportion of nodes for which the correct output was predicted (node accuracy), the average proportion of orbits which have entirely correct predictions (orbit accuracy), and the proportion of graphs which were entirely solved (graph accuracy). Since we are dealing with orbit-equivariant functions, accuracy is calculated by collecting the orbits of the graph and computing the multiset intersections of model and target outputs for each orbit.

**Results**  Our main results are shown in Table 1 and full results are shown in Appendix C.5. In the plots, "max_orbit_gcn" refers to $m$-Orbit-Transform-GCN. Since we found that slightly better results were obtained for Bioisostere when standard cross-entropy is used, those results are shown, with orbit-sorting cross-entropy used for the other two datasets. Across all 3 datasets and metrics, Orbit-Indiv-GCNs consistently and definitively exhibited the best performance. GCNs exhibited the worst performance on Bioisostere and would have achieved an accuracy of 0 on the Alchemy-Max-Orbit datasets, highlighting the need for models that go beyond equivariance. Unique-ID-GCNs consistently outperformed RNI-GCNs, which aligns with the experiments of Morris et al. (2022) on non-equivariant problems. It is also interesting to note that, whilst RNI-GCNs achieved the 3rd highest test accuracy on Bioisostere, they had the highest accuracy on the training set, suggesting that the model is more inclined to overfit.

On the Alchemy-Max-Orbit-2 dataset, 2-Orbit-Transform-GCNs achieved comparable performance to Orbit-Indiv-GCNs, whilst the other two models performed poorly. This suggests that the transformation model can be useful on certain tasks, when the max-orbit of the function is low. In contrast, on the Alchemy-Max-Orbit-6 dataset, 6-Orbit-Transform-GCNs had by far the worst performance among all the models, demonstrating that the transformation becomes infeasible for large max-orbit sizes. This is expected, since the number of output channels required for the transformation is linear in the max-orbit. Overall, whilst Alchemy-Max-Orbit-2 has smaller max-orbits than Alchemy-Max-Orbit-6, the models tend to perform better on the latter dataset because there are fewer graphs in Alchemy that possess a maximum orbit of 6 than those that possess a maximum orbit of 2, meaning that similar structures appear repeatedly when the Max-Orbit-6 graphs are augmented to extend the dataset.

Model performance on the test datasets during training are shown in Figure 6 for the experiments using cross-entropy loss, and in Figure 7 for those using orbit-sorting cross-entropy loss. In addition to Orbit-Indiv-GCNs performing the best out of all the models, they also converge by far the fastest. We attribute this to their strong structural inductive bias: they are exactly orbit-equivariant whilst not deviating much from the provenly effective architecture of GCNs. Abboud et al. (2021); Morris et al. (2022) found that RNI-GCNs can be slow to converge, but on Bioisostere, we find them to converge comparatively swiftly. A comparison between Figures 6 and 7 highlights the importance of using orbit-sorting cross-entropy loss. The performance of Orbit-Transform-GCNs is mostly unaffected, since they are trained as equivariant models: this is an advantage unique to Orbit-Transform-GCNs. The models also do not show improvement on Bioisostere when using orbit-sorting cross-entropy,

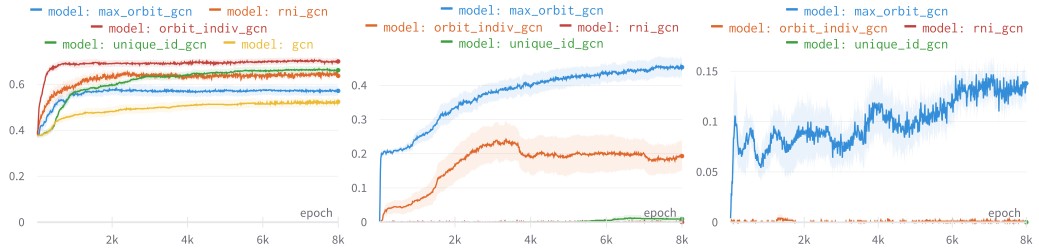

Figure 6: Graph accuracy with standard error on the test datasets across all models using cross-entropy loss: Bioisostere (left), Alchemy-Max-Orbit-2 (center), and Alchemy-Max-Orbit-6 (right).

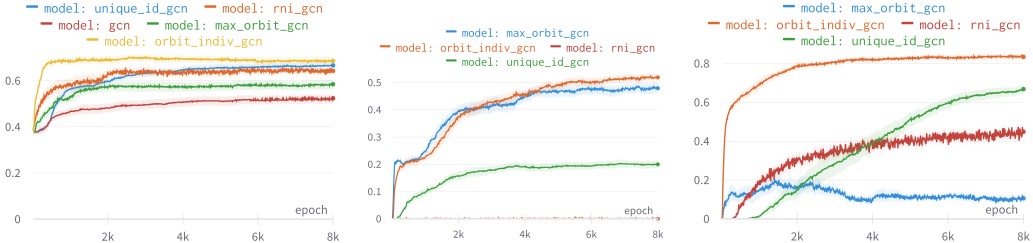

Figure 7: Graph accuracy with standard error on the test datasets across all models using orbit-sorting cross-entropy: Bioisostere (left), Alchemy-Max-Orbit-2 (center), and Alchemy-Max-Orbit-6 (right).

which is probably because each node from an orbit that is to be swapped out is chosen deterministically to be the one with the lowest index. This is not the case for the Max-Orbit datasets.

## 6 CONCLUSION

We show the limits of equivariant functions when it comes to producing different outputs for similar nodes. We define a less restrictive property, orbit-equivariance, to address this issue while still retaining a strong structural inductive bias and show where it lies in the hierarchy of node-labelling functions. We also provide a brief taxonomy of orbit-equivariant functions using a further property, `max-orbit`. We present four different ways to extend GNNs to solve non-equivariant problems and analyze their expressivity. We evaluate these models on both real-world and synthetic datasets, finding that one of our novel models (Orbit-Indiv-GNN) exhibits the best performance in all scenarios. We also find that the use of a specialist loss function can be necessary for the models to train.

The novelty of orbit-equivariance as a property provides considerable scope for future work in this area. Firstly, the idea of orbit-equivariance can be generalized to other data structures besides graphs (Bronstein et al., 2021) and related to node embeddings (Srinivasan & Ribeiro, 2019). GNNs have also proven to be useful for aiding in designing optimal bioisosteres, but this is a complex task and deserves further investigation. Moreover, the establishment of orbit-equivariance as a formal property could lead to the identification of other problems that require non-equivariant and orbit-equivariant models to solve. Finally, there may be ways to design better, specially tailored orbit-equivariant GNNs to solve such problems.

**Limitations** Our work considers only graphs where the nodes are not individualized; this does not account for application domains where the nodes have noisy features that are almost identical, but not precisely. Our theorems focus on property satisfaction and there are open questions as to how expressive the new proposed models are. The scope of our experiments is restricted to datasets where non-equivariant functions are required. Furthermore, the Bioisostere dataset does not capture all aspects of actual drug design, since we only optimize for a single molecular property instead of many, and do not consider biological effects.

## 7 ETHICS STATEMENT

We see no ethical concerns in our work. This is a largely theoretical paper, and we see no potential negative societal impact of the theories or of the particular applications. Our models and datasets do not have a high risk for misuse, so we are releasing them to the public. Furthermore, we did not use human subjects and did not need IRB approval. All papers and repositories have been cited and licenses explicitly mentioned for the software used.

## 8 REPRODUCIBILITY STATEMENT

Code, data, and instructions for how to reproduce our experimental results are given in the supplemental material. Full model hyperparameters (including random seeds) are given in the appendix. Important training details are given in section 5 of the main paper, and full details are given in the appendices. Full details on the amount and type of compute used are given in the appendix. All terminology and symbols are explicitly defined, and the assumptions are clearly stated across the paper, particularly in sections 3 and 4. Complete proofs are given in the appendices and intuitions given in the main paper.

ACKNOWLEDGMENTS

Matthew Morris is funded by an EPSRC scholarship (CS2122_EPSRC_1339049). This work was also supported by the SIRIUS Centre for Scalable Data Access (Research Council of Norway, project 237889), Samsung Research UK, the EPSRC projects AnaLOG (EP/P025943/1), OASIS (EP/S032347/1), UKFIRES (EP/S019111/1) and ConCur (EP/V050869/1). The authors would like to acknowledge the use of the University of Oxford Advanced Research Computing (ARC) facility in carrying out this work http://dx.doi.org/10.5281/zenodo.22558, as well as thank Ismail Ceylan for helpful early-stage discussions.

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

# A PROOFS

## A.1 PROPOSITION 1

**Proposition.** *Let $f$ be an equivariant node-labelling function and let $G$ be a labelled graph in its domain. If $v, w \in V(G)$ are similar, then $f(G)_v = f(G)_w$.*

*Proof.* Define $\sigma$ to be a permutation that maps $v \rightarrow w$, and maps other nodes such that $\sigma$ is an automorphism of $G$. Such a permutation exists since $v$ and $w$ are similar. Since $\sigma$ is an automorphism, $\sigma \cdot G = G$. So we prove:

$$
\begin{aligned}
f(G)_v &= (\sigma^{-1} \cdot \sigma \cdot f(G))_v && \text{(identity permutation)} \\
&= (\sigma^{-1} \cdot f(\sigma \cdot G))_v && \text{(equivariance)} \\
&= f(\sigma \cdot G)_w && (\sigma^{-1} \text{ maps } w \rightarrow v) \\
&= f(G)_w && (\sigma \text{ is an automorphism})
\end{aligned}
$$

$\square$

## A.2 PROPOSITION 2

**Proposition.** *All equivariant functions are orbit-equivariant, but not vice-versa. There exist node-labelling functions which are not orbit-equivariant.*

*Proof.* We prove the three claims of this proposition separately.

### All equivariant functions are orbit-equivariant.

Let a node-labelling function $f$ on domain $D$ be equivariant. Then for all labelled graphs $G$ and permutations $\sigma$ on $V(G)$, $f(\sigma \cdot G) = \sigma \cdot f(G)$. To prove that $f$ is orbit-equivariant, choose arbitrary labelled graph $G \in D$, permutation $\sigma$ on $V(G)$, and let $r \in R(G)$ be any orbit of $G$. Then,

$$
\begin{aligned}
\{\!\!\{ f(\sigma \cdot G)_{\sigma(v)} \mid v \in r \}\!\!\} &= \{\!\!\{ (\sigma^{-1} \cdot f(\sigma \cdot G))_v \mid v \in r \}\!\!\} && \text{(permute vector)} \\
&= \{\!\!\{ (\sigma^{-1} \cdot \sigma \cdot f(G))_v \mid v \in r \}\!\!\} && \text{(equivariance)} \\
&= \{\!\!\{ f(G)_v \mid v \in r \}\!\!\} && \text{(inverse permutation)}
\end{aligned}
$$

Thus, $f$ is orbit-equivariant.

### There are orbit-equivariant functions which are not equivariant.

For an example of an orbit-equivariant function that is not equivariant, consider the node-labelling function $f$, defined on the set of all labelled graphs, that maps input graphs $G_1, G_2, G_3$ as shown, and maps each other graph $G$ to $\{0\}^{|G|}$.

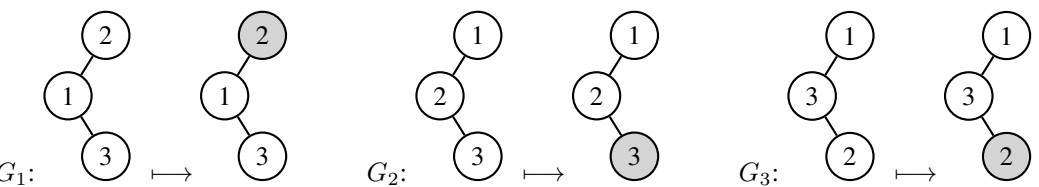

This function is orbit-equivariant. There are two orbits of $G_1$, $r_1 := \{1\}, r_2 = \{2, 3\}$. Consider the permutation $\sigma = \{(1, 2), (2, 1), (3, 3)\}$. Note that $\sigma \cdot G_1 = G_2$ and $\sigma \cdot G_2 = G_1$. For $r_1$, we have $\{\!\!\{ f(\sigma \cdot G_1)_{\sigma(v)} \mid v \in r_1 \}\!\!\} = \{\!\!\{ f(G_1)_v \mid v \in r_1 \}\!\!\} = \{\!\!\{ 0 \}\!\!\}$, and for $r_2$, we have $\{\!\!\{ f(\sigma \cdot G_1)_{\sigma(v)} \mid v \in r_2 \}\!\!\} = \{\!\!\{ f(G_1)_v \mid v \in r_2 \}\!\!\} = \{\!\!\{ 0, 1 \}\!\!\}$.

Any permutation on a graph in $D = \{G_1, G_2, G_3\}$ yields a graph in $D$, and the only graphs that permute to a graph in $D$ are the graphs in $D$. In other words, $D$ and $\mathcal{G} \setminus D$ are both closed under permutation, where $\mathcal{G}$ is the set of all labelled graphs. Each graph $G \in D$ has two orbits $r_1, r_2$, of size 1 and 2 respectively, with $\{\!\{f(G)_v \mid v \in r_1\}\!\} = \{\!\{0\}\!\}$ and $\{\!\{f(G)_v \mid v \in r_2\}\!\} = \{\!\{0, 1\}\!\}$. For $G_1, r_1 = \{1\}, r_2 = \{2, 3\}$, for $G_2, r_1 = \{2\}, r_2 = \{1, 3\}$, and for $G_3, r_1 = \{3\}, r_2 = \{1, 2\}$.

Thus, following the same unfolding of the definition as above, we see that $\forall G \in D$, $\forall$ permutations $\sigma$ on $V(G)$, for the corresponding orbit $r_1$ of $G$, we have $\{\!\{f(\sigma \cdot G)_{\sigma(v)} \mid v \in r_1\}\!\} = \{\!\{f(G)_v \mid v \in r_1\}\!\} = \{\!\{0\}\!\}$, and for $r_2$, we have $\{\!\{f(\sigma \cdot G)_{\sigma(v)} \mid v \in r_2\}\!\} = \{\!\{f(G)_v \mid v \in r_2\}\!\} = \{\!\{0, 1\}\!\}$.

Furthermore, each graph $G \in \mathcal{G} \setminus D$ has $f(G) = \{0\}^{|V(G)|}$, and for any permutation $\sigma$ on $V(G)$, $\sigma \cdot G \in \mathcal{G} \setminus D$, so $f(\sigma \cdot G) = \{0\}^{|V(G)|}$. Thus, $\forall G \in \mathcal{G} \setminus D$, $\forall$ permutations $\sigma$ on $V(G)$, $\forall r \in R(G)$, we have $\{\!\{f(\sigma \cdot G)_{\sigma(v)} \mid v \in r\}\!\} = \{\!\{f(G)_v \mid v \in r\}\!\} = \{\!\{0\}\!\}^{|r|}$.

Thus, $f$ is orbit-equivariant, since the definition holds $\forall G \in D$ and $\forall G \in \mathcal{G} \setminus D$.

However, $f$ is not equivariant. For the same permutation $\sigma = \{(1, 2), (2, 1), (3, 3)\}$, $f(\sigma \cdot G_1) = f(G_2) = (0, 0, 1)$ and $\sigma \cdot f(G_1) = \sigma \cdot (0, 1, 0) = (1, 0, 0)$. So $f(\sigma \cdot G_1) \neq \sigma \cdot f(G_1)$.

### There exist node-labelling functions which are not orbit-equivariant.

Consider a node-labelling function $f$ that exhibits the following behavior for distinct graphs $G_1, G_2, G_3$ and assigns all other graphs $G$ to $\{0\}^{|G|}$.

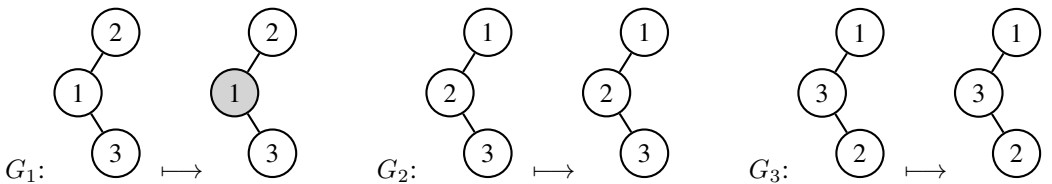

This function is not orbit-equivariant. Consider permutation $\sigma = \{(1, 2), (2, 1), (3, 3)\}$, for which $\sigma \cdot G_1 = G_2$, and consider orbit $r = \{1\}$ of $G_1$. Then $\{\!\{f(\sigma \cdot G_1)_{\sigma(v)} \mid v \in r\}\!\} = \{\!\{f(G_2)_2\}\!\} = \{\!\{0\}\!\}$ and $\{\!\{f(G_1)_v \mid v \in r\}\!\} = \{\!\{f(G_1)_1\}\!\} = \{\!\{1\}\!\}$. So there exists an orbit $r \in R(G_1)$ and permutation $\sigma$ such that $\{\!\{f(\sigma \cdot G_1)_{\sigma(v)} \mid v \in r\}\!\} \neq \{\!\{f(G_1)_v \mid v \in r\}\!\}$.

$\square$

## A.3 PROPOSITION 3

**Proposition.** *If $f$ is orbit-equivariant and* `max-orbit`$(f) = 1$*, then $f$ is equivariant.*

*Proof.* Let $G$ be a labelled graph in the domain $D$ of $f$ and $\sigma$ a permutation on $V(G)$. RTP: $f(\sigma \cdot G) = \sigma \cdot f(G)$.

Since `max-orbit`$(f) = 1$, the maximum across all orbits $r \in R(G)$ of the number of unique values in $\{\!\{f(G)_v \mid v \in r\}\!\}$ is 1. Thus, $\forall r \in R(G)$, $\forall v, w \in r$, $f(G)_v = f(G)_w$.

Since $f$ is orbit-equivariant:

$$\forall \text{ orbits } r \in R(G), \ \{\!\{f(\sigma \cdot G)_{\sigma(v)} \mid v \in r\}\!\} = \{\!\{f(G)_v \mid v \in r\}\!\}$$

$$\Rightarrow \quad \forall \text{ orbits } r \in R(G), \ \forall v \in r, \ f(\sigma \cdot G)_{\sigma(v)} = f(G)_v \qquad \text{(RHS multiset all equal)}$$

$$\Rightarrow \quad \forall v \in V(G), \ f(\sigma \cdot G)_{\sigma(v)} = f(G)_v \qquad \text{(all orbits = all nodes)}$$

$$\Rightarrow \quad \forall v \in V(G), \ (\sigma^{-1} \cdot f(\sigma \cdot G))_v = f(G)_v \qquad \text{(permute vector)}$$

$$\Rightarrow \quad \forall v \in V(G), \ (\sigma \cdot \sigma^{-1} \cdot f(\sigma \cdot G))_v = (\sigma \cdot f(G))_v$$

$$\Rightarrow \quad \forall v \in V(G), \ f(\sigma \cdot G)_v = (\sigma \cdot f(G))_v \qquad \text{(inverse permutation)}$$

$$\Rightarrow \quad f(\sigma \cdot G) = \sigma \cdot f(G)$$

$\square$

## A.4 Theorem 1

**Theorem.** *If orbit-1-WL assigns different colourings to $u, v \in V(G)$, then $u, v$ are not similar.*

*Proof.* We prove this by induction on $n$, the number of iterations the orbit-1-WL algorithm has taken. The orbit-1-WL algorithm is given formally in Appendix B.1. The stated theorem thus becomes: "if orbit-1-WL assigns different colourings to nodes $u, v \in G$ after $n$ iterations, then they are not similar".

### Base case: n=0

If $u, v$ have different colourings after no iterations of orbit-1-WL, then they have different labels in the original graph. This implies that $u$ and $v$ are not similar, since any automorphism must be label-preserving.

### Inductive step: assume true for n

Let $C_n : V(G) \to C$ be the colouring of $G$ after $n$ iterations of orbit-1-WL and similarly $C_{n+1}$ after $n+1$ iterations. We have nodes $u, v \in G$ s.t. $C_{n+1}(u) \neq C_{n+1}(v)$, and aim to prove that $u$ and $v$ are not similar.

### Case 1: $C_n(u) \neq C_n(v)$

Then as assumed in the inductive step, $u$ and $v$ are not similar.

### Case 2: $C_n(u) = C_n(v)$

$C_{n+1}(u)$ is computed by $C_{n+1}(u) := h(C_n(u), \{C_n(w) \mid w \in N(u)\})$, where $h$ is the WL hash function. But then since $C_n(u) = C_n(v)$ and $C_{n+1}(u) \neq C_{n+1}(v)$, $\{C_n(w) \mid w \in N(u)\} \neq \{C_n(w) \mid w \in N(v)\}$.

If $|N(u)| \neq |N(v)|$, then $u$ and $v$ are trivially not similar, since any automorphism must be edge-preserving. So assume that $|N(u)| = |N(v)|$. The above set inequality thus implies that there does not exist a bijection $\pi : N(u) \to N(v)$ such that $\forall w \in N(u), C_n(w) = C_n(\pi(w))$.

### Assume to the contrary that $u$ and $v$ are similar.

Then there exists an automorphism $\sigma : G \to G$ such that $\sigma(u) = v$. This implies that $\forall w_1 \in N(u) \exists w_2 \in N(v)$ such that $\sigma(w_1) = w_2$, otherwise $\sigma$ is not edge-preserving. So we have $|N(u)|$ pairs $(w_1, w_2)$ of similar nodes, where each $w_1 \in N(u)$, $w_2 \in N(v)$. Define $V_\sigma := \{(w_1, w_2) \mid w_1 \in N(u), w_2 := \sigma(w_1)\}$ to be the set containing those pairs.

If for some $(w_1, w_2) \in V_\sigma$, $C_n(w_1) \neq C_n(w_2)$, then $w_1$ and $w_2$ are not similar by assumption in the inductive step, which creates a contradiction.

So instead assume that $\forall (w_1, w_2) \in V_\sigma$, $C_n(w_1) = C_n(w_2)$. This implies that $\forall w \in N(u), C_n(w) = C_n(\sigma(w))$. Thus, when restricted to $N(u)$, $\sigma$ induces a bijection $\pi : N(u) \to N(v)$ such that $\forall w \in N(u), C_n(w) = C_n(\pi(w))$. But this is a contradiction, since it was shown above that no such bijection can exist.

### Conclusion

Thus, either way, by contradiction, we can conclude that $u$ and $v$ are not similar. This completes the inductive step, as we have shown that the hypothesis holds for $C_{n+1}$. Thus, the theorem is true $\forall n$ by induction.

$\square$

A.5    THEOREM 2

**Theorem.** *The expressivity of our proposed models is as follows, where for a class of models $X$, the claim "$X$ is not equivariant" means that "there exists $f \in X$ such that $f$ is not equivariant":*

1. *Unique-ID-GNNs are not orbit-equivariant and, for any fixed $n$, can approximate any node-labelling function $f : G_n \to \mathbb{R}^n$, where $G_n$ is the set of graphs with $\le n$ nodes.*

2. *RNI-GNNs are equivariant in expectation and, for any fixed $n$, can approximate any equivariant function $f : G_n \to \mathbb{R}^n$ with probability arbitrarily close to 1 (Abboud et al., 2021; Morris et al., 2022). They can approximate some non-equivariant and orbit-equivariant functions with probability arbitrarily close to 1, but there exist RNI-GNNs which, with probability arbitrarily close to 1, are not orbit-equivariant.*

3. *Orbit-Indiv-GNNs are not equivariant but are orbit-equivariant on graphs whose orbits are distinguishable by orbit-1-WL. For any $m \in \mathbb{Z}^+$, there exist Orbit-Indiv-GNNs $f$ with* `max-orbit(f) > m`.

4. *$m$-Orbit-Transform-GNNs $f$ are not equivariant but are orbit-equivariant on graphs whose orbits are distinguishable by orbit-1-WL. They have* `max-orbit(f) ≤ m` *and there exist $m$-Orbit-Transform-GNNs $f$ with* `max-orbit(f) = m`.

*Proof.* The theorem follows from the four following Lemmas (1, 2, 3, and 4), each of which proves the expressivity of one of the models in the Theorem.    □

**Lemma 1.** *Unique-ID-GNNs are not orbit-equivariant and, for any fixed $n$, can approximate any node-labelling function $f : G_n \to \mathbb{R}^n$, where $G_n$ is the set of graphs with $\le n$ nodes.*

*Proof.* This proof has two sections, one for each of its claims.

### Unique-ID-GNNs are not orbit-equivariant.

The following is a trivial example of a Unique-ID-GNN $f$ that is not orbit-equivariant. Let $f$ be a Unique-ID-GNN defined by $f(G) = f_{\text{GNN}}(f_{\text{unique}}(G))$, where $f_{\text{unique}}(G)_v := (v, G_v)$ deterministically appends a unique label (the unique label is $v$ itself, since it is already an integer) to each node $v$. Define $f_{\text{GNN}}$ as a GNN with a single layer, where $f_{\text{aggr}}^{\sigma_1}$ and $f_{\text{read}}^{\sigma_1}$ are arbitrary and $f_{\text{update}}^{\sigma_1}(\boldsymbol{\lambda}_v^m, \_, \_) := \boldsymbol{\lambda}_v^m$. The GNN does not do any actual message-passing, and simply keeps the original label. Then, consider the below graphs $G_1, G_2$, where the nodes have empty labels.

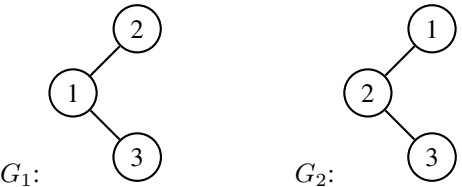

Consider the permutation $\sigma = \{(1, 2), (2, 1), (3, 3)\}$, for which $\sigma \cdot G_1 = G_2$, and consider the orbit $r = \{1\}$ of $G_1$. Then $\{\!\!\{ f(\sigma \cdot G_1)_{\sigma(v)} \mid v \in r \}\!\!\} = \{\!\!\{ f(G_2)_2 \}\!\!\} = \{\!\!\{ 2 \}\!\!\}$ and $\{\!\!\{ f(G_1)_v \mid v \in r \}\!\!\} = \{\!\!\{ f(G_1)_1 \}\!\!\} = \{\!\!\{ 1 \}\!\!\}$. So there exists an orbit $r \in R(G_1)$ and permutation $\sigma$ such that $\{\!\!\{ f(\sigma \cdot G_1)_{\sigma(v)} \mid v \in r \}\!\!\} \neq \{\!\!\{ f(G_1)_v \mid v \in r \}\!\!\}$. Hence, the Unique-ID-GNN $f$ is not orbit-equivariant.

### Unique-ID-GNNs can, for any fixed $n$, approximate any node-labelling function $f : G_n \to \mathbb{R}^n$.

We aim to prove that, for any node-labelling function $f : G_n \to \mathbb{R}^n$, there exists a Unique-ID-GNN that approximates $f$ with arbitrary precision $\epsilon$. For this proof, we rely on Theorem 4 of Morris et al. (2022), which we state here:

*Let $n \geq 1$ and $f : G_n \to \mathbb{R}^n$ be equivariant. Then $\forall \epsilon > 0$, there exists a GNN with unique node IDs that $\epsilon$-approximates $f$.*

Note that their definition of a "GNN with unique node IDs" is identical to Unique-ID-GNN. $G_n$ is defined as the set of all graphs with at most $n$ nodes, so $n$ can be made arbitrarily large so as to include any particular graph. They also use a codomain of $\mathbb{R}$ for each node feature, but note that the result extends to real vectors of arbitrary size. In their proof, they show that there exists a Unique-ID-GNN $g(f_{\text{unique}}(G))$ that with arbitrary precision yields a final label of $\boldsymbol{\lambda}_v = (u_v, r_v)$ for each node $v$, where $r_v$ uniquely identifies the graph $G$ in $G_n$ (up to isomorphism) and $u_v$ is the original unique ID given to $v$.

However, this only identifies the graph up to isomorphism. So, we first prove that for any graphs $G, H$ with $V(G) = V(H)$, $f_{\text{unique}}(G) \equiv f_{\text{unique}}(H) \iff G = H$. If $G = H$, then trivially $f_{\text{unique}}(G) = f_{\text{unique}}(H)$ and thus $f_{\text{unique}}(G) \equiv f_{\text{unique}}(H)$. If instead $f_{\text{unique}}(G) \equiv f_{\text{unique}}(H)$, then there exists a permutation $\sigma$ on $V(G)$ such that $\sigma \cdot f_{\text{unique}}(G) = f_{\text{unique}}(H)$. The permutation maps each $v \in V(G)$ (with unique ID $u_v$) to the vertex $w \in V(H)$ with the same unique ID, since the permutation is label-preserving. But since $f_{\text{unique}}$ is deterministic, $v = w$. Thus, $\forall v, w \in V(G)$, $(v, w) \in E(G) \iff (\sigma(v), \sigma(w)) \in E(H) \iff (v, w) \in E(H)$, and $G_v = H_w$. So $G = H$. This proves that identifying $f_{\text{unique}}(G)$ up to isomorphism is equivalent to identifying $G$ up to equality.

Thus, define a GNN $f_{\text{copy}}$ with a single layer that simply copies the unique label: $f_{\text{aggr}}^{\sigma_1}$ and $f_{\text{read}}^{\sigma_1}$ are arbitrary and $f_{\text{update}}^{\sigma_1}((v, G_v), \_, \_) := (v, v, G_v),$. The GNN does not do any actual message-passing, and simply copies the unique ID portion of the original label. But then the GNN $g(f_{\text{copy}}(f_{\text{unique}}))$ yields a final label of $\boldsymbol{\lambda}_v = (u_v, r_v)$ for each node $v$, where $r_v$ uniquely identifies the graph $G$ (up to isomorphism) and $u_v$ is the original unique ID given to $v$. But since $r_v$ identifies $f_{\text{unique}}(G)$ up to isomorphism, it identifies $G$ up to equality.

But then we can define a mapping $f^{\text{node}}$ such that for each labelled graph $G$, $v \in G$, $f^{\text{node}}(u_v, r_v) = f(G)_v$. This mapping can be approximated by an MLP with arbitrary precision and appended onto the GNN $g(f_{\text{copy}}(f_{\text{unique}}))$ as a final update layer. This yields a GNN that approximates $f$ with arbitrary precision. $\qquad\square$

**Lemma 2.** *RNI-GNNs are equivariant in expectation and, for any fixed $n$, can approximate any equivariant function $f : G_n \to \mathbb{R}^n$ with probability arbitrarily close to 1 (Abboud et al., 2021; Morris et al., 2022). They can approximate some non-equivariant and orbit-equivariant functions with probability arbitrarily close to 1, but there exist RNI-GNNs which, with probability arbitrarily close to 1, are not orbit-equivariant.*

*Proof.* This proof has three sections, one for each of its claims.

### RNI-GNNs are equivariant in expectation and, for any fixed $n$, can approximate any equivariant function $f : G_n \to \mathbb{R}^n$ with probability arbitrarily close to 1.

Note that since RNI-GNNs are functions that include randomness, we can no longer discuss them simply approximating functions. Instead, we need to refer to them approximating functions with a given probability.

To formally define "with probability arbitrarily close to 1": Abboud et al. (2021) say that a randomized function $X$ that associates with every graph $G \in G_n$ a random variable $X(G)$ is an $(\epsilon, \delta)$-*approximation* of $f$ if for all $G \in G_n$ it holds that $\Pr(|f(G) - X(G)| \le \epsilon) \ge 1 - \delta$. Note that an RNI-GNN $N$ computes such functions $X$. If $X$ is computed by $N$, we say that $N$ $(\epsilon, \delta)$-*approximates* $f$. If for all $\epsilon, \delta > 0$, there is a RNI-GNN that $(\epsilon, \delta)$-*approximates* $f$, then we abbreviate this by saying that the RNI-GNN "approximates $f$ with probability arbitrarily close to 1".

Abboud et al. (2021) prove that RNI-GNNs are equivariant in expectation and that they can approximate any invariant graph function with probability arbitrarily close to 1 (in their Theorem 1). Morris et al. (2022) extend the universality results of Abboud et al. (2021) and prove that they can approximate any equivariant node-labelling function with probability arbitrarily close to 1 (in their Theorem 2).

### RNI-GNNs can approximate some non-equivariant and orbit-equivariant functions with probability arbitrarily close to 1.

We provide an example of a non-equivariant and orbit-equivariant function that RNI-GNNs can approximate with probability arbitrarily close to 1. This is done to demonstrate their ability to go beyond equivariance and into orbit-equivariance, but it remains an open question as to whether RNI-GNNs can approximate any orbit-equivariant function.

Define a function $f$ that relabels the nodes of each input graph as follows and returns the relabelled graph. For a labelled graph $G$, $f(G)_v = r_{G,v}$, where $r_{G,v}$ is some value that uniquely identifies the orbit of $v$ in $G$ up to isomorphism. Formally: $r_{G,v} = r_{H,w} \iff$ there exists an isomorphism $\alpha : G \to H$ (the graphs containing nodes $v$ and $w$, respectively) such that $\alpha(v) = w$. Thus, for any permutation $\sigma$, $r_{G,v} = r_{\sigma \cdot G, \sigma(v)}$, since $\sigma$ is an isomorphism from $G$ to $\sigma \cdot G$.

This function $f$ is equivariant, since for all labelled graphs $G$, all permutations $\sigma$ on $V(G)$ and all nodes $v \in V(G)$, we have $f(\sigma \cdot G)_v = r_{\sigma \cdot G, v} = r_{G, \sigma^{-1}(v)} = f(G)_{\sigma^{-1}(v)} = \sigma \cdot f(G)_v$. Since $f$ is equivariant, Theorem 2 of Morris et al. (2022) proves that it can be approximated with probability arbitrarily close to 1 by an RNI-GNN $g$.

Now consider another such function $f_{\text{copy}}$ that relabels the nodes of each input graph and returns the relabelled graph, such that $f_{\text{copy}}(G)_v = (\eta_v, r_{G,v})$. This is the same as $f$, except it also copies the random noise $\eta_v$ given to node $v$ in the input graph $G$. $f_{\text{copy}}$ is also equivariant, as a trivial consequence of $f$ being equivariant, and can be approximating by extending the RNI-GNN $g$ to copy the random noise for each node.

Now fix some graph $H$ and node $w \in H$ such that the orbit of $w$ has size at least 2. Then consider a node-labelling function $f_{\text{max}}$, such that $f_{\text{max}}(G)_v = 1$ if $r_{G,v} = r_{H,w}$ and $\eta_v$ is the maximum of $\eta_u$ for all $u \in V(G)$ such that $r_{G,u} = r_{H,w}$. Otherwise, $f_{\text{max}}(G)_v = 0$.

The node-labelling function $h$ defined by $h(G) := f_{\text{max}}(f_{\text{copy}}(G))$ can be approximated by extending the previously mentioned RNI-GNN $g$. Append one final layer to the GNN, defined by $f_{\text{read}}(\{\!\!\{(\eta_u, r_{G,u}) \mid u \in V(G)\}\!\!\}) := \max(\{\!\!\{\eta_u \mid u \in V(G) \text{ and } r_{G,u} = r_{H,w}\}\!\!\})$. This function is permutation-invariant and so is a valid choice for $f_{\text{read}}$. $f_{\text{update}}$ is defined by

$f_{\text{update}}((\eta_v, r_{G,v}), \_, \eta_u) := 1$ if $r_{G,v} = r_{H,w}$ and $\eta_v = \eta_u$, and $:= 0$ otherwise. This extension approximates $h$ with probability arbitrarily close to 1, since a unique value in $\{\!\!\{\eta_u \mid u \in V(G) \text{ and } r_{G,u} = r_{H,w}\}\!\!\}$ is maximal with probability arbitrarily close to 1, because the random values are sampled from a continuous interval.

It now remains to be shown that the node-labelling function $h$ is orbit-equivariant. To prove that $h$ is orbit-equivariant, let $G$ be a labelled graph, $\sigma$ be a permutation on $V(G)$, and $r \in R(G)$ an orbit.

**Case 1:** $r_{\sigma \cdot G, \sigma(u)} = r_{H,w}$ for some node $u \in r$. Then:

$$\{\!\!\{ h(\sigma \cdot G)_{\sigma(v)} \mid v \in r \}\!\!\} = \{\!\!\{ f_{\text{max}}(f_{\text{copy}}(\sigma \cdot G))_{\sigma(v)} \mid v \in r \}\!\!\} = \{\!\!\{1\}\!\!\} \cup \{\!\!\{0\}\!\!\}^{|r|-1}$$

Also, since $r_{G,u} = r_{\sigma \cdot G, \sigma(u)}$ (proven when $f$ is defined), we can derive that $r_{G,u} = r_{H,w}$. Thus:

$$\{\!\!\{ h(G)_v \mid v \in r \}\!\!\} = \{\!\!\{ f_{\text{max}}(f_{\text{copy}}(G))_v \mid v \in r \}\!\!\} = \{\!\!\{1\}\!\!\} \cup \{\!\!\{0\}\!\!\}^{|r|-1}$$

**Case 2:** $r_{\sigma \cdot G, \sigma(u)} \neq r_{H,w}$ for every node $u \in r$. Then:

$$\{\!\!\{ h(\sigma \cdot G)_{\sigma(v)} \mid v \in r \}\!\!\} = \{\!\!\{ f_{\text{max}}(f_{\text{copy}}(\sigma \cdot G))_{\sigma(v)} \mid v \in r \}\!\!\} = \{\!\!\{0\}\!\!\}^{|r|}$$

Again, since $r_{G,u} = r_{\sigma \cdot G, \sigma(u)}$ for every $u \in V(G)$, we can derive that $r_{G,u} \neq r_{H,w}$ for every $u \in r$. Thus:

$$\{\!\!\{ h(G)_v \mid v \in r \}\!\!\} = \{\!\!\{ f_{\text{max}}(f_{\text{copy}}(G))_v \mid v \in r \}\!\!\} = \{\!\!\{0\}\!\!\}^{|r|}$$

Hence, in either case, $\{\!\!\{ h(\sigma \cdot G)_{\sigma(v)} \mid v \in r \}\!\!\} = \{\!\!\{ h(G)_v \mid v \in r \}\!\!\}$, so $h$ is orbit-equivariant. However, it is non-equivariant, since $\{\!\!\{ h(G)_v \mid v \in r \}\!\!\} = \{\!\!\{1\}\!\!\} \cup \{\!\!\{0\}\!\!\}^{|r|-1}$ when $G = H$ and $w \in r$ (note that $|r| \geq 2$ by the definition of $w$), which shows that the function produces different values for similar nodes, violating the property of equivariance proved in Proposition 1.

**There exist RNI-GNNs which, with probability arbitrarily close to 1, are not orbit-equivariant.**

Consider an RNI-GNN $f$ with a single layer. The readout function $f_{\text{read}}^{\theta_1''}$ is defined by $f_{\text{read}}^{\theta_1''}(\{\!\!\{ (\eta_w, \boldsymbol{\lambda}_w^0) \mid w \in V(G) \}\!\!\}) := \max(\{\!\!\{ \eta_w \mid w \in V(G) \}\!\!\})$, where $\eta_w$ is the random noise given to node $w$. The update function $f_{\text{update}}^{\theta_1}$, is defined by $f_{\text{update}}^{\theta_1}((\eta_v, \boldsymbol{\lambda}_v^1), \_, \eta_{\text{max}}) := 1$ if $\eta_v = \eta_{\text{max}}$, and 0 otherwise, where $\eta_{\text{max}}$ is the output of $f_{\text{read}}^{\theta_1''}$.

Thus, for a graph $G$ with $n$ nodes, $f(G)_w = 1$ for some $w \in V(G)$ (chosen uniformly at random), and $f(G)_u = 0$ for all $u \in V(G)$ such that $u \neq w$.

As $n$ becomes large for graphs with small orbit sizes, the probability that $f(G)_w = 1$ and $f(G)_u = 1$ for any $w, u \in V(G)$ such that $w, u$ are in the same orbit of $G$, across two separate applications of $f$, becomes arbitrarily small.

Thus also, the probability that for all labelled graphs $G$, permutations $\sigma$ on $V(G)$, and orbits $r \in R(G)$, it holds that $\{\!\!\{ f(\sigma \cdot G)_{\sigma(v)} \mid v \in r \}\!\!\} = \{\!\!\{ f(G)_v \mid v \in r \}\!\!\}$ becomes arbitrarily small. So the RNI-GNN $f$ is not orbit-equivariant with probability arbitrarily close to 1. $\qquad \square$

**Lemma 3.** *Orbit-Indiv-GNNs are not equivariant but are orbit-equivariant on graphs whose orbits are distinguishable by orbit-1-WL. For any $m \in \mathbb{Z}^+$, there exists Orbit-Indiv-GNNs $f$ with* `max-orbit`$(f) > m$.

*Proof.* This proof has four sections, one for each of its claims and one demonstrating the consequences of having an input graph whose orbits are not distinguishable by orbit-1-WL.

### Orbit-Indiv-GNNs are not equivariant.

We provide a simple example of an Orbit-Indiv-GNN $f$ that is not equivariant, defined on a graph $G$ by $f(G) := f_{\text{mlp}}(f_{\text{orbit\_indiv}}(f_{\text{GNN}}(G)))$. $f_{\text{GNN}}$ is defined by $f_{\text{GNN}}(G)_v := 0$. Define $f_{\text{mlp}}$ to be the identity function. Let $G$ be the below graph.

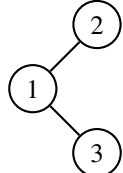

Then $f(G) = ((1,0),(1,0),(2,0))$. Nodes 2 and 3 are similar in $G$. But $f(G)_2 = (1,0)$ and $f(G)_2 = (2,0)$, breaking the property of equivariance proved in Proposition 1. Thus, Orbit-Indiv-GNN $f$ is not equivariant.

### For any $m \in \mathbb{Z}^+$, there exist Orbit-Indiv-GNNs $f$ with `max-orbit`$(f) > m$.

Let $m \in \mathbb{Z}^+$ and $D = \{C_n \mid n \in \mathbb{Z}^+\}$ be the set of all single-cycle graphs. Let $f$ be an Orbit-Indiv-GNN defined on $D$ by $f(G) := f_{\text{mlp}}(f_{\text{orbit\_indiv}}(f_{\text{GNN}}(G)))$. $f_{\text{GNN}}$ is defined by $f_{\text{GNN}}(G)_v := 0$. Define $f_{\text{mlp}}$ to be the identity function.

Then for any $G = C_n \in D$, orbit-1-WL correctly identifies all nodes as belonging to the same orbit, so $f(G) = ((1,0),(2,0),...,(n,0))$. This has $n$ different outputs within the same orbit. But $C_{m+1} \in D$, so `max-orbit`$(f) \geq m + 1 > m$.

### Orbit-Indiv-GNNs are orbit-equivariant on graphs whose orbits are distinguishable by orbit-1-WL.

Let $f$ be an Orbit-Indiv-GNN defined by $f(G) := f_{\text{mlp}}(f_{\text{orbit\_indiv}}(f_{\text{GNN}}(G)))$ and $D$ a set of all graphs whose orbits are distinguishable by orbit-1-WL. Then since $\forall r \in R(G)$, $v, w \in r$, $f_{\text{GNN}}(G)_v = f_{\text{GNN}}(G)_w$, we have, for all labelled graphs $G \in D$, permutations $\sigma$ on $V(G)$, and orbits $r \in R(G)$:

$$\{\!\!\{ f(G)_v \mid v \in r \}\!\!\} = \{\!\!\{ f_{\text{mlp}}((u, f_{\text{GNN}}(G)_v)) \mid v \in r, \ u \in \{1, 2, ..., |r|\} \}\!\!\}$$

Also, $\forall r \in R(G)$, $v, w \in r$, $f_{\text{GNN}}(\sigma \cdot G)_{\sigma(v)} = f_{\text{GNN}}(\sigma \cdot G)_{\sigma(w)}$, since $v, w$ similar in $G$ implies that $\sigma(v), \sigma(w)$ are similar in $\sigma \cdot G$. So we have, for all labelled graphs $G$, permutations $\sigma$ on $V(G)$, and orbits $r \in R(G)$:

$$
\begin{aligned}
&\{\!\!\{ f(\sigma \cdot G)_{\sigma(v)} \mid v \in r \}\!\!\} \\
=& \{\!\!\{ f_{\text{mlp}}((u, f_{\text{GNN}}(\sigma \cdot G)_{\sigma(v)})) \mid v \in r, \ u \in \{1, 2, ..., |r|\} \}\!\!\} && \text{(since } f_{\text{mlp}} \text{ is element-wise)} \\
=& \{\!\!\{ f_{\text{mlp}}((u, (\sigma \cdot f_{\text{GNN}}(G))_{\sigma(v)})) \mid v \in r, \ u \in \{1, 2, ..., |r|\} \}\!\!\} && \text{(since } f_{\text{GNN}} \text{ is equivariant)} \\
=& \{\!\!\{ f_{\text{mlp}}((u, f_{\text{GNN}}(G)_v)) \mid v \in r, \ u \in \{1, 2, ..., |r|\} \}\!\!\}
\end{aligned}
$$

Thus, $\{\!\!\{ f(G)_v \mid v \in r \}\!\!\} = \{\!\!\{ f(\sigma \cdot G)_{\sigma(v)} \mid v \in r \}\!\!\}$, so Orbit-Indiv-GNN $f$ is orbit-equivariant.

**The expressivity of Orbit-Indiv-GNNs for distinguishing nodes into different orbits is limited by orbit-1-WL.**

First, note that $f_{\text{GNN}}$ has the same expressive power as orbit-1-WL for distinguishing nodes into different orbits and that both $f_{\text{GNN}}$ and orbit-1-WL are explicitly used in computing the output of an Orbit-Indiv-GNN. Thus, Orbit-Indiv-GNNs are limited by the expressivity of orbit-1-WL. We will briefly unpack the consequences of this here.

The contrapositive of Theorem 1 is: if $v, w \in V(G)$ are similar, then orbit-1-WL assigns the same colorings to $u, v$. Thus, the following statement (used in the above proof that Orbit-Indiv-GNNs are orbit-equivariant) still holds true, despite the limited expressive power of $f_{\text{GNN}}$: $\forall r \in R(G)$, $v, w \in r$, $f_{\text{GNN}}(G)_v = f_{\text{GNN}}(G)_w$.

In the above proof, it was assumed that for every orbit $r \in R(G)$, orbit-1-WL correctly identifies the nodes in $r$ and thus that $f_{\text{orbit\_indiv}}$ distributes the labels $\{1, 2, ..., |r|\}$ among the nodes of the orbit. However, consider the following graph for which orbit-1-WL fails to identify the orbits:

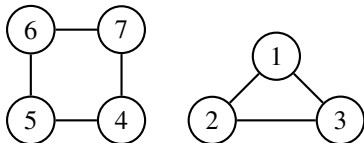

The actual orbits are $r_1 = \{1, 2, 3\}$, $r_2 = \{4, 5, 6, 7\}$, but orbit-1-WL incorrectly identifies them as $r_3 = \{1, 2, 3, 4, 5, 6, 7\}$. Consider Orbit-Indiv-GNN $f$, defined on a graph $G$ by $f(G) := f_{\text{mlp}}(f_{\text{orbit\_indiv}}(f_{\text{GNN}}(G)))$. $f_{\text{GNN}}$ is defined by $f_{\text{GNN}}(G)_v := 0$. Define $f_{\text{mlp}}$ to be the identity function. Then $f(G) = ((1, 0), (2, 0), (3, 0), (4, 0), (5, 0), (6, 0), (7, 0))$. Let $\sigma$ be the permutation $\{(1, 7), (7, 1), (2, 2), (3, 3), (4, 4), (5, 5), (6, 6)\}$. Then $\sigma \cdot G$ is the following graph:

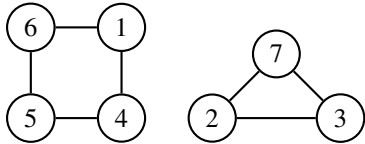

Then $\{\!\{ f(G)_v \mid v \in r_1 \}\!\} = \{\!\{ (1, 0), (2, 0), (3, 0) \}\!\}$, but since $f_{\text{orbit\_indiv}}$ uses the orbit-1-WL orbits, $\{\!\{ f(\sigma \cdot G)_{\sigma(v)} \mid v \in r_1 \}\!\} = \{\!\{ (7, 0), (2, 0), (3, 0) \}\!\}$. So $f$ is not orbit-equivariant for this graph, due to the expressivity of Orbit-Indiv-GNNs for distinguishing nodes into different orbits being limited by orbit-1-WL. The same problem occurs on any input graph where the orbits cannot be correctly identified by orbit-1-WL. This theoretical issue can be alleviated by using a more expressive (and likely slower) orbit-identification algorithm than orbit-1-WL, or by only operating on graphs whose nodes can be distinguished by orbit-1-WL. $\qquad\square$

**Lemma 4.** *$m$-Orbit-Transform-GNNs $f$ are not equivariant but are orbit-equivariant on graphs whose orbits are distinguishable by orbit-1-WL. They have* `max-orbit`$(f) \leq m$ *and there exist $m$-Orbit-Transform-GNNs $f$ with* `max-orbit`$(f) = m$.

*Proof.* This proof has four sections, one for each of its claims and one demonstrating the consequences of having an input graph whose orbits are not distinguishable by orbit-1-WL.

### $m$-Orbit-Transform-GNNs are not equivariant.

We provide a simple example of a 2-Orbit-Transform-GNN $f$ that is not equivariant, defined on a graph $G$ by $f(G) := f_{\text{transform}}(f_{\text{GNN}}(G))$. In this case, $o := 1$ (the number of final output channels). $f_{\text{GNN}}$ is defined as stated below for input graphs $G_1, G_2, G_3$, and maps each other graph $G$ to $\{((0, 99), (0, 99))\}^{|G|}$. In the diagram below, the colors of nodes represent their features, with *white* $= ((0, 99), (0, 99))$ and *gray* $= ((1, -99), (2, -99))$. The large final channel (99) represents not swapping from the default (i.e. if the final channel is the largest, then the default should not be swapped out for the value in this partition).

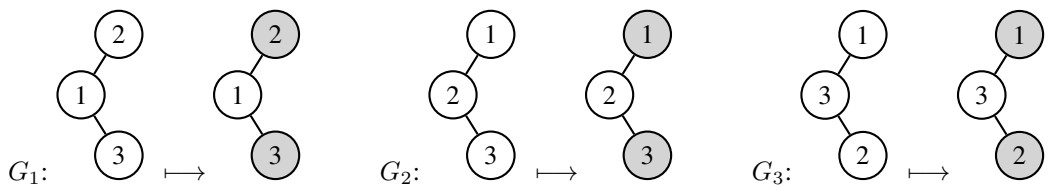

Then $f(G_1) = (0, 1, 2)$. Nodes 2 and 3 are similar in $G_1$. But $f(G_1)_2 = 1$ and $f(G_1)_2 = 2$, breaking the property of equivariance proved in Proposition 1. Thus, 2-Orbit-Transform-GNN $f$ is not equivariant. A similar example for any $m > 2$ shows that $m$-Orbit-Transform-GNNs are not equivariant.

### $m$-Orbit-Transform-GNNs $f$ have `max-orbit`$(f) \leq m$ and there exist $m$-Orbit-Transform-GNNs $f$ with `max-orbit`$(f) = m$.

Since we aim to convey the `max-orbit` of an entire class of models instead of just a single function, we prove that there exist specific $m$-Orbit-Transform-GNNs $f$ that have `max-orbit`$(f) = m$ and that there are no $m$-Orbit-Transform-GNNs $f$ with `max-orbit`$(f) > m$. Note that there do exist trivial $m$-Orbit-Transform-GNNs $f$ such that `max-orbit`$(f) < m$.

We prove this in two parts. First, we prove that for any $m$-Orbit-Transform-GNN $f$, `max-orbit`$(f) \leq m$. This follows from there being exactly $m$ partitions in the output of $f_{\text{GNN}}(G)$. For each graph $G$ and orbit $r \in R(G)$, step (1) of $f_{\text{transform}}$ gives a single label to all nodes in the orbit. Then step (2) adds a new label to the orbit at most $\min(m, |r|) - 1$ times, meaning there are at most $m$ different labels in the orbit after step (2) is finished.

Second, we give an example of an $m$-Orbit-Transform-GNN $f$ that has `max-orbit`$(f) = m$. $f_{\text{GNN}}$ is defined by $f(G)_v = ((1, -99), (2, -99), ..., (m, -99))$ for each node $v \in V(G)$. Consider the cycle graph $H := C_m$ (a single cycle of $m$ nodes). It has a single orbit, $r = \{1, 2, ..., m\}$, which is identifiable by orbit-1-WL. Thus, $f(H) = (1, 2, ..., m)$, so there are $m$ unique values in $\{\!\{ f(H)_v \mid v \in r \}\!\}$.

### $m$-Orbit-Transform-GNNs are orbit-equivariant on graphs whose orbits are distinguishable by orbit-1-WL.

Let $f$ be a $m$-Orbit-Transform-GNN defined by $f(G) := f_{\text{transform}}(f_{\text{GNN}}(G))$ and $D$ a set of all graphs whose orbits are distinguishable by orbit-1-WL. Then since $\forall r \in R(G)$, $v, w \in r$, $f_{\text{GNN}}(G)_v = f_{\text{GNN}}(G)_w$, we have, for all labelled graphs $G \in D$, permutations $\sigma$ on $V(G)$, and orbits $r \in R(G)$:

$$\{\!\{ f(G)_v \mid v \in r \}\!\} = f_{\text{orbit\_transform}}(f_{\text{GNN}}(G)_w) \quad \text{(for any } w \in r)$$

where $f_{\text{orbit\_transform}}$ is a function that yields the multiset of outputs of $f_{\text{transform}}$ for the orbit containing $w$. These outputs are all set to a single value by step (1) of $f_{\text{transform}}$, and are then changed by each iteration of step (2). Also, $\forall r \in R(G)$, $v, w \in r$, $f_{\text{GNN}}(\sigma \cdot G)_{\sigma(v)} = f_{\text{GNN}}(\sigma \cdot G)_{\sigma(w)}$, since $v, w$ similar in $G$ implies that $\sigma(v), \sigma(w)$ are similar in $\sigma \cdot G$. So we have, for all labelled graphs $G$, permutations $\sigma$ on $V(G)$, and orbits $r \in R(G)$:

$$\{\!\{f(\sigma \cdot G)_{\sigma(v)} \mid v \in r\}\!\} = f_{\text{orbit\_transform}}(f_{\text{GNN}}(\sigma \cdot G)_{\sigma(w)}) \qquad \text{(for any } w \in r)$$
$$= f_{\text{orbit\_transform}}((\sigma \cdot f_{\text{GNN}}(G))_{\sigma(w)}) \qquad \text{(since } f_{\text{GNN}} \text{ is equivariant)}$$
$$= f_{\text{orbit\_transform}}(f_{\text{GNN}}(G)_w)$$

Thus, $\{\!\{f(G)_v \mid v \in r\}\!\} = \{\!\{f(\sigma \cdot G)_{\sigma(v)} \mid v \in r\}\!\}$, so $m$-Orbit-Transform-GNN $f$ is orbit-equivariant.

### The expressivity of $m$-Orbit-Transform-GNNs for distinguishing nodes into different orbits is limited by orbit-1-WL.

First, note that $f_{\text{GNN}}$ has the same expressive power as orbit-1-WL for distinguishing nodes into different orbits and that both $f_{\text{GNN}}$ and orbit-1-WL are explicitly used in computing the output of a $m$-Orbit-Transform-GNN. Thus, $m$-Orbit-Transform-GNNs are limited by the expressivity of orbit-1-WL. We will briefly unpack the consequences of this here.

The contrapositive of Theorem 1 is: if $v, w \in V(G)$ are similar, then orbit-1-WL assigns the same colorings to $u, v$. Thus, the following statement (used in the above proof that $m$-Orbit-Transform-GNNs are orbit-equivariant) still holds true, despite the limited expressive power of $f_{\text{GNN}}$: $\forall r \in R(G)$, $v, w \in r$, $f_{\text{GNN}}(G)_v = f_{\text{GNN}}(G)_w$.

In the above proof, it was assumed that for every orbit $r \in R(G)$, orbit-1-WL correctly identifies the nodes in $r$ and thus that $f_{\text{orbit\_transform}}$ yields the multiset of outputs of $f_{\text{transform}}$ for the orbit containing $w$. However, consider the following graph for which orbit-1-WL fails to identify the orbits:

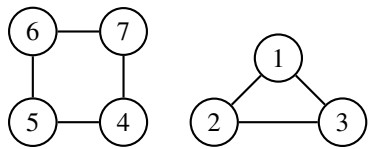

The actual orbits are $r_1 = \{1, 2, 3\}$, $r_2 = \{4, 5, 6, 7\}$, but orbit-1-WL incorrectly identifies them as $r_3 = \{1, 2, 3, 4, 5, 6, 7\}$. Consider 7-Orbit-Transform-GNN $f$, defined on a graph $G$ by $f(G) := f_{\text{transform}}(f_{\text{GNN}}(G))$. $f_{\text{GNN}}$ is defined by $f_{\text{GNN}}(G)_v := ((1, -99), (2, -99), (3, -99), (4, -99), (5, -99), (6, -99), (7, -99))$. Then $f(G) = (1, 2, 3, 4, 5, 6, 7)$. Let $\sigma$ be the permutation $\{(1, 7), (7, 1), (2, 2), (3, 3), (4, 4), (5, 5), (6, 6)\}$. Then $\sigma \cdot G$ is the following graph:

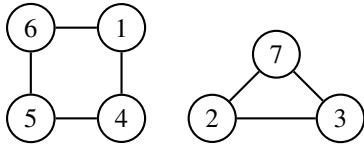

Then $\{\!\{f(G)_v \mid v \in r_1\}\!\} = \{\!\{1, 2, 3\}\!\}$, but since $f_{\text{transform}}$ uses the orbit-1-WL orbits, $\{\!\{f(\sigma \cdot G)_{\sigma(v)} \mid v \in r_1\}\!\} = \{\!\{7, 2, 3\}\!\}$. So $f$ is not orbit-equivariant for this graph, due to the expressivity of $m$-Orbit-Transform-GNNs for distinguishing nodes into different orbits being limited by orbit-1-WL. The same problem occurs on any input graph where the orbits cannot be correctly identified by orbit-1-WL. This theoretical issue can be alleviated by using a more expressive (and likely slower) orbit-identification algorithm than orbit-1-WL, or by only operating on graphs whose nodes can be distinguished by orbit-1-WL. $\qquad\square$

## B  Orbit-1-WL

### B.1  Definition

The 1-WL algorithm (Weisfeiler & Leman, 1968) can be formalized algorithmically as follows:

1. Initialize $C_0 : V(G) \to C$ by $C_0(v) = c_1$ for each node $v \in V(G)$, where $C$ is a codomain of colors and $c_1 \in C$. $C_0$ is a node coloring function at iteration 0 of the algorithm.

2. At each iteration $n$ of the algorithm (starting from $n = 0$), we define a new coloring function $C_{n+1}$. For each node $v \in V(G)$, set $C_{n+1}(v) := h(C_n(v), \{\!\{C_n(w) \mid w \in N(v)\}\!\})$, where $h$ is a hash function and $N(v)$ is the set of all neighbors of $v$ in $G$.

3. At each iteration $n$ of the algorithm, also compute a partition $P_{n+1}$ of $V(G)$, where for any nodes $v, w \in V(G)$, $v, w \in Q$ for any $Q \in P$ if and only if $C_{n+1}(v) = C_{n+1}(w)$. In other words, the nodes are partitioned by their colors.

4. If at any point, $P_{n+1} = P_n$, the algorithm halts and returns $h(\{\!\{C_n(w) \mid w \in V(G)\}\!\})$.

Orbit-1-WL operates in the exact same way, except that when the algorithm halts, it returns $P_n$ instead of $h(\{\!\{C_n(w) \mid w \in V(G)\}\!\})$. $P_n$ represents a partition of nodes into their orbits by orbit-1-WL.

### B.2  Successful Examples

As an example, here is orbit-1-WL identifying the orbits of a graph:

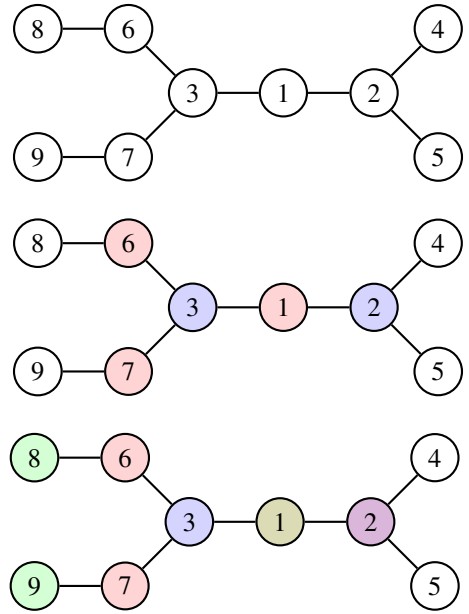

This yields orbits of $\{\{1\}, \{2\}, \{3\}, \{4, 5\}, \{6, 7\}, \{8, 9\}\}$, so orbit-1-WL has correctly identified the orbits. As another example, consider the following two graphs, $G_1$ and $G_2$:

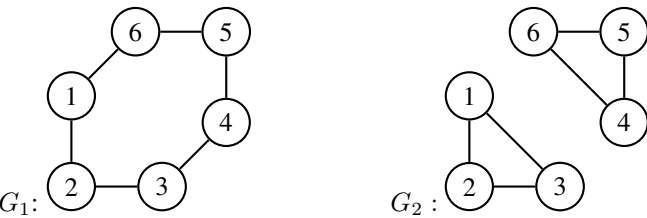

Despite 1-WL being unable to distinguish between these two graphs, orbit-1-WL correctly identifies the orbits for both (all nodes are in the same orbit). This illustrates why it is important to distinguish between 1-WL and orbit-1-WL, since there are cases where 1-WL fails and orbit-1-WL succeeds, and conversely there are cases where orbit-1-WL fails and 1-WL succeeds (as will be shown in a later section).

### B.3 COUNTEREXAMPLES FOR COMPLETENESS

Theorem 1 proves the soundness of orbit-1-WL. However, orbit-1-WL is not complete. That is: there exists a graph $G$ and nodes $v, w \in G$ such that orbit-1-WL assigns the same colorings to $v$ and $w$, but $v$ and $w$ are not similar. We show several such graphs here.

In the following example, orbit-1-WL colors all the nodes the same (since the graph is 2-regular), so it identifies all nodes as being in the same orbit. However, the true orbits are $\{\{1, 2, 3\}, \{4, 5, 6, 7\}\}$.

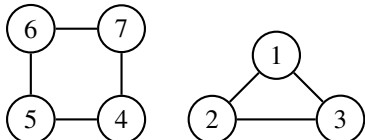

The above example can be scaled up into infinitely many other 2-regular graphs for which the same problem arises, yielding infinitely many counterexamples. It is worth noting as well that if a graph $G$ contains a graph $H$ as a maximally connected sub-graph (i.e. a collection of components of $G$), such that the orbits of $H$ are incorrectly identified by orbit-1-WL, then the orbits of $G$ are also incorrectly identified by orbit-1-WL. This yields another way to generate infinitely many counterexamples.

Here follows two other non-trivial 3-regular graphs (one planar and one non-planar) for which orbit-1-WL fails. Orbit-1-WL colors all nodes the same since the graphs are 3-regular. The first has actual orbits of $\{\{1, 2, 7, 8\}, \{3, 4, 5, 6\}\}$. The second has actual orbits of $\{\{1, 2, 7, 8, 9, 10\}, \{3, 4, 5, 6\}\}$.

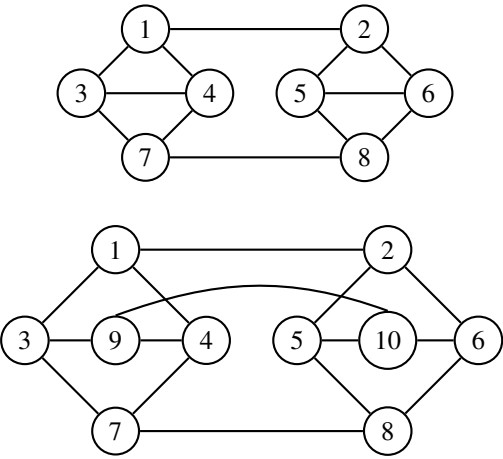

Here follows a counterexample (found in the Alchemy dataset (Chen et al., 2019)) which is not a regular graph. It is colored by orbit-1-WL as shown.

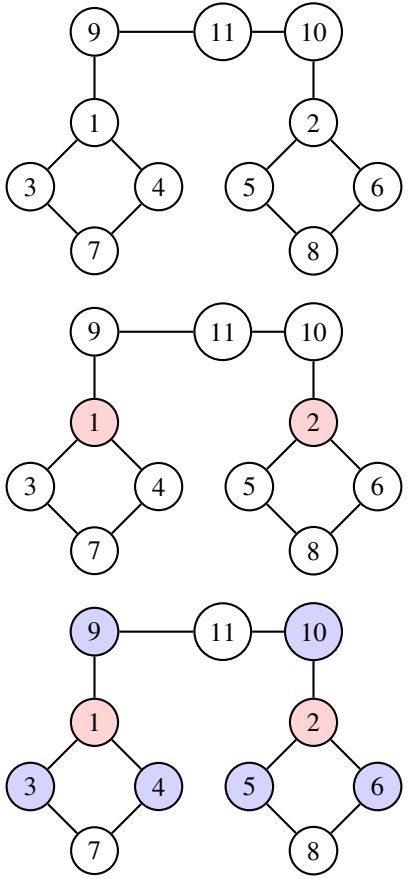

However, the true orbits of the above graph are

$$\{\{1, 2\}, \{3, 4, 5, 6\}, \{7, 8\}, \{9, 10\}, \{11\}\}$$

Out of the 202579 graphs in Alchemy (Chen et al., 2019), orbit-1-WL only predicts the incorrect orbits for 5. We calculated this by comparing the true orbits and orbit-1-WL orbits for every graph in the dataset.

## B.4 ORBIT-k-WL

Similar to classic WL (Morris et al., 2019), there is a strict hierarchy of orbit-WL algorithms: orbit-1-WL, orbit-3-WL, orbit-4-WL, ..., such that for all $k \geq 3$ there are graphs for which orbit-$k$-WL will correctly identify the orbits but orbit-$(k$-1)-WL will not. The below example shows a construction that proves this:

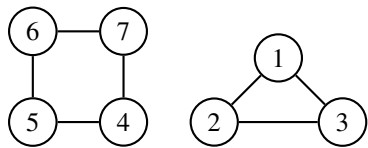

Neither orbit-1-WL or orbit-2-WL correctly identifies the orbits of the graph, but orbit-3-WL does. For any $k \geq 3$, the graph consisting of a $(k + 1)$-cycle and a $k$-cycle has its orbits correctly identified by orbit-$k$-WL but not by orbit-$(k$-1)-WL.

Table 2: Properties of the new datasets proposed in the paper.

| Dataset | | **Bioisostere** | **Alchemy-Max-Orbit-**2 | **Alchemy-Max-Orbit-**6 |
|---|---|---|---|---|
| Graphs | | 961 | 1000 | 1000 |
| Input Features | | 43 | 6 | 6 |
| Possible Targets per Node | | 44 | 2 | 6 |
| Nodes | Average | 10.3 | 10 | 12.1 |
| | Min | 2 | 10 | 9 |
| | Max | 15 | 10 | 22 |
| Edges | Average | 10.1 | 10.4 | 12.5 |
| | Min | 1 | 9 | 8 |
| | Max | 17 | 12 | 22 |

## C  EXPERIMENTS

### C.1  COMPUTATIONAL COMPLEXITY

With respect to the computational complexity of our proposed models, Unique-ID-GNNs and RNI-GNNs have no extra overhead, besides yielding a larger input vector for each label in the GNN. Orbit-Indiv-GNNs use 1-WL, which is worst-case $O(n)$, where $n$ is the number of nodes in the graph. Appending unique IDs and the application of the final MLP are also $O(n)$. So overall, the additional complexity of Orbit-Indiv-GNNs is $O(n)$.

Likewise, for $m$-Orbit-Transform-GNNs, 1-WL is worst-case $O(n)$. The output size of the GNN is larger than before: it is now $m(o + 1)$ instead of $o$, which adds additional computation to the GNN. Apart from this, there is no additional computational cost during training. When making predictions, the transform is applied, which is worst-case $O(n)$. So overall, the additional complexity of $m$-Orbit-Transform-GNNs when making predictions is $O(n)$.

### C.2  DATASETS

In Table 2, we present a summary of the new datasets we propose in this paper. Full details are provided in the following sections.

#### C.2.1  BIOISOSTERE

The task in **Bioisostere** is, given an input molecule as a graph, output a label for each node which represents either keeping the atom or swapping it out for some other particular atom, in such a way that the final resulting molecule has minimal lipophilicity; as discussed in Section 1, this is an important factor in drug design. Solving this task requires a non-equivariant function when an atom being swapped out comes from an orbit of size $\geq 2$.

To construct this dataset, we first retrieved 961 small drug molecules from ChEMBL (Mendez et al., 2019). The version of ChEMBL we used is CHEMBL32 and the content is licensed under the CC Attribution-ShareAlike 3.0 Unported license. We filtered to only include drug molecules. From there, we filtered by "Type" to include only those with the type "Small molecule", and further filtered by "Molecular Weight" to only include molecules with a molecular weight $\leq 199$. This yielded a CSV file of molecules, which was cleaned up into `chembl.smi` a file of SMILES code and molecule name pairs.

We then used MMPDB (RDKit, 2023a) to compute many different bioisosteres for each molecule that each differ by at most one atom from the original molecule. `mmpdb fragment chembl.smi -o chembl.fragments --num-cuts 1` was used to compute the fragments of all the molecules from CHEMBL that can be created by cutting a single non-ring bond. `mmpdb index chembl.fragments -o chembl.mmpdb` was then used to group fragments with the same R-groups into a database. For each molecule with SMILE code "input_smile", `mmpdb transform --smiles input_smile chembl.mmpdb` was used to compute a list of bioisosteres of the molecule. For each of those bioisosteres (and the original molecule), we used RDKit (RDKit, 2023b)

to compute the lipophilicity and selected the one that yielded the lowest value as the target for each molecule. Both MMPDB and RDKit are licensed under the BSD 3-Clause License.

Of these molecules, only 588 actually require some swapping out of an atom to achieve minimal lipophilicity, and only 156 require a non-equivariant function. All 961 molecules are used in the dataset. The function $f$ required to solve the task is orbit-equivariant with `max-orbit`$(f) = 2$.

### C.2.2 ALCHEMY-MAX-ORBIT

An **Alchemy-Max-Orbit-**$m$ dataset is constructed using Alchemy (Chen et al., 2019; Morris et al., 2020), a large dataset of organic molecules. The dataset constructor is given $m$ as a hyperparameter. First, it removes all duplicate graphs from Alchemy using 1-WL as a heuristic. Next, all graphs without an orbit of size $\geq m$ are removed, where the orbits are computed using orbit-1-WL.[2] If the number of graphs remaining is larger than the desired size of the constructed dataset (1000 for this paper), then graphs are dropped from the end of the dataset to make the dataset the desired size.

If the number of graphs remaining is instead smaller than the desired size of the constructed dataset, then new graphs are added to the dataset using two graph generators. Both generators make small changes to graphs already in the dataset, broadly preserving the structures whilst adding new non-isomorphic graphs. The constructor continues to alternate between applying each generator, with the process immediately halting when the dataset has attained the desired size. Each generator is applied to every graph that is in the dataset so far. However, before any new graph is added to the dataset, it is checked using its 1-WL hash to ensure that a graph isomorphic to it is not already in the dataset. For each graph given to it, the first generator takes every orbit of the graph, and varies all the node features within the orbit in the same way. The second generator takes every orbit of the input graph and, for every node $v$ within the orbit, appends a new node $w$ to the graph that is connected to only $v$.

The task in the dataset is to identify the largest orbit(s) in the graph (where the orbits are computed by orbit-1-WL) and, for each largest orbit $r$, distribute the labels $T := \{\!\!\{1, 1, ..., 1, 1, 2, ..., m\}\!\!\}$ at random without replacement amongst the orbit's nodes, where $|T| = |r|$. Thus, $T$ contains the multiset $\{\!\!\{1, 2, ..., m\}\!\!\}$ and is padded with 1s to ensure that there are the same number of labels in it as nodes in the orbit. Every node not part of a largest orbit should be assigned the label of $0$. Equivariant functions such as GNNs cannot solve any of the examples in the dataset. The function $f$ required to solve the task is orbit-equivariant with `max-orbit`$(f) = m$.

### C.3 HYPERPARAMETERS

Fixed hyperparameters across all experiments are shown in Table 3. The fixed hyperparameters were chosen by preliminary optimization experiments of GCN on Bioisostere. Hyperparameters for the experiments on Bioisostere are shown in Table 4. Hyperparameters for the experiments on Alchemy-Max-Orbit-2 are shown in Table 5. Hyperparameters for the experiments on Alchemy-Max-Orbit-6 are shown in Table 6.

### C.4 LOSS

Here follows a full explanation of *orbit-sorting cross-entropy loss*. Let $f$ be a GNN and $D$ a dataset of graphs. Let $t$ be a "target function" that yields a category for each node in the input graph. For each orbit $r$ of the graph $G$ (computed by orbit-1-WL), collect the model outputs $(f(G)_v \mid v \in r)$ and the target outputs $(t(G)_v \mid v \in r)$ for the orbit, both as tuples. The category predictions of the model are $(\arg\_\max(f(G)_v) \mid v \in r)$, since each output channel of $f(G)_v$ corresponds to a category. Fix some a-priori order on the output categories and define a function `sort` that takes in a tuple of categories $c$ and returns the permutation $\sigma$ that, when applied to the indices of the tuple, yields the tuple sorted by category. For example, given the tuple $c = (8, 2, 4, 7, 3)$ of categories, `sort`$(c) = \sigma = \{(1, 5), (2, 1), (3, 3), (4, 4), (5, 2)\}$, and $\sigma \cdot c = (2, 3, 4, 7, 8)$.

---

[2]We elected to use orbit-1-WL to compute the orbits when creating the Alchemy-Max-Orbit datasets, because some of the graphs in the dataset were too large to efficiently compute all automorphisms and thus the orbits. For consistency, we elected to compute all orbits using orbit-1-WL, rather than exactly computing the orbits of the graphs that were small enough and using orbit-1-WL for the others.

Table 3: Fixed model parameters for all experiments

| Group | Parameter | Value | Description |
|---|---|---|---|
| Model | gnn_layers | 4 | Number of message-passing layers in the GNN |
| | gnn_hidden_size | 40 | Size of each hidden layer in the GNN |
| | rni_channels | 10 | Number of channels of random noise to append |
| | use_cpu | True | Force use of the CPU over CUDA |
| Evaluation | train_eval_interval | 10 | Eval on train set interval in epochs |
| | test_eval_interval | 10 | Eval on test set interval in epochs |
| | loss_log_interval | 10 | How often to log the model loss in epochs |
| Dataset | train_on_entire_dataset | False | Whether to tain on the entire dataset |
| | train_split | 0.9 | Ratio of dataset to use for the train set |
| | bioisostere_only_equivariant | False | Filter out non-equivariant examples from Bioisostere |
| | shuffle_targets_in_max_orbit | True | Shuffle targets within orbits of Alchemy-Max-Orbit |
| | shuffle_dataset | True | Shuffle dataset before splitting into train / test sets |
| Training | learning_rate | 0.0001 | Adam optimiser learning rate |
| | weight_decay | 5e-4 | Adam optimiser weight decay |
| | n_epochs | 8000 | Number of epochs (full training set) to run |

Table 4: Model parameters for Bioisostere experiments

| Parameter | Value | Description |
|---|---|---|
| dataset | bioisostere | Dataset |
| loss | [ cross_entropy, orbit_sorting_cross_entropy ] | Loss function |
| model | [ gcn, unique_id_gcn, rni_gcn, orbit_indiv_gcn, max_orbit_gcn ] | Model (max_orbit = Orbit-Transform-GNN) |
| model_max_orbit | 2 | Max orbit for Orbit-Transform-GNN |
| seed | [1,2,3,4,5,6,7,8,9,10] | Seed for all random generators |

Table 5: Model parameters for Alchemy-Max-Orbit-2 experiments

| Parameter | Value | Description |
|---|---|---|
| dataset | alchemy | Base dataset |
| max_orbit_alchemy | 2 | Max-orbit size ($m$) in Alchemy-Max-Orbit-$m$ |
| loss | [ cross_entropy, orbit_sorting_cross_entropy ] | Loss function |
| model | [ unique_id_gcn, rni_gcn, orbit_indiv_gcn, max_orbit_gcn ] | Model (max_orbit = Orbit-Transform-GNN) |
| model_max_orbit | 2 | Max orbit for Orbit-Transform-GNN |
| seed | [1,2,3,4,5,6,7,8,9,10] | Seed for all random generators |

Table 6: Model parameters for Alchemy-Max-Orbit-6 experiments

| Parameter | Value | Description |
|---|---|---|
| dataset | alchemy | Base dataset |
| max_orbit_alchemy | 6 | Max-orbit size ($m$) in Alchemy-Max-Orbit-$m$ |
| loss | [ cross_entropy, orbit_sorting_cross_entropy ] | Loss function |
| model | [ unique_id_gcn, rni_gcn, orbit_indiv_gcn, max_orbit_gcn ] | Model (max_orbit = Orbit-Transform-GNN) |
| model_max_orbit | 6 | Max orbit for Orbit-Transform-GNN |
| seed | [1,2,3,4,5,6,7,8,9,10] | Seed for all random generators |

For the entire graph, initially set $T := t(G)$. Then, for each orbit $r$ (computed by orbit-1-WL), we compute $\sigma_f := \texttt{sort}((\arg\_\max(f(G)_v) \mid v \in r))$ and $\sigma_t := \texttt{sort}((t(G)_v \mid v \in r))$, the permutations that sort the model category predictions and the targets respectively. We then compute a re-ordered target $T^r$ for the orbit, with $T^r := \sigma_f^{-1} \cdot (\sigma_t \cdot (t(G)_v \mid v \in r))$. Then $T[r]$ is set to $T^r$, where $T[r]$ accesses the indices of $T$ using the nodes in $r$.

Once all orbits have been iterated through, we compute the cross-entropy between the new targets $T$ and the model outputs $(f(G)_v \mid v \in r)$.

To better illustrate the mechanisms of orbit-sorting cross-entropy loss, here follows an example. For an input graph $G$, $f(G)$ (converted to categorical outputs with arg_max, for ease of notation) and $t(G)$ are shown below, where colors represent node features. White = 0, red = 1, green = 2, and blue = 3.

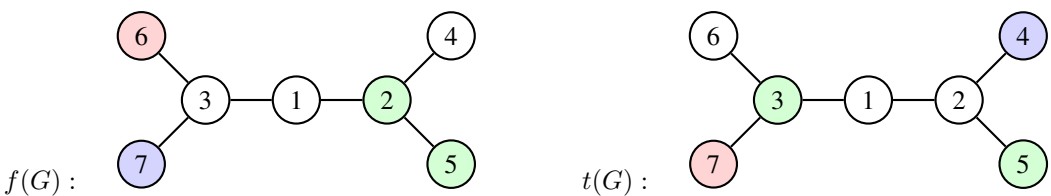

The orbits of this graph are $r_1 = \{1\}, r_2 = \{2, 3\}, r_3 = \{4, 5, 6, 7\}$, as computed correctly by orbit-1-WL. Set $T := t(G) = (0, 0, 2, 3, 2, 0, 1)$ initially. $r_1$ has size 1, so no re-ordering is possible. For $r_2$, $(\arg\_\max(f(G)_v) \mid v \in r_2) = (2, 0)$, so $\sigma_f = \{(1, 2), (2, 1)\}$. Also, $(t(G)_v \mid v \in r_2) = (0, 2)$, so $\sigma_t = \{(1, 1), (2, 2)\}$. Thus $T^{r_2} := \sigma_f^{-1} \cdot (\sigma_t \cdot (t(G)_v \mid v \in r_2)) = \sigma_f^{-1} \cdot (\sigma_t \cdot (0, 2)) = (2, 0)$. $T[r_2]$ is then set to $T^{r_2}$, so $T = (0, \mathbf{2}, \mathbf{0}, 3, 2, 0, 1)$ (the affected part of the tuple is highlighted in bold).

For $r_3$, $(\arg\_\max(f(G)_v) \mid v \in r_3) = (0, 2, 1, 3)$, so $\sigma_f = \{(1, 1), (2, 3), (3, 2), (4, 4)\}$. Also, $(t(G)_v \mid v \in r_3) = (3, 2, 0, 1)$, so $\sigma_t = \{(1, 4), (2, 3), (3, 1), (4, 2)\}$. Thus $T^{r_3} := \sigma_f^{-1} \cdot (\sigma_t \cdot (t(G)_v \mid v \in r_3)) = \sigma_f^{-1} \cdot (\sigma_t \cdot (3, 2, 0, 1)) = (0, 2, 1, 3)$. $T[r_3]$ is then set to $T^{r_3}$, so $T = (0, 2, 0, \mathbf{0}, \mathbf{2}, \mathbf{1}, \mathbf{3})$. Cross-entropy in then computed between the new targets $T$ and the model outputs $(f(G)_v \mid v \in r) = (0, 2, 0, 0, 2, 1, 3)$.

For the $m$-Orbit-Transform-GNN models, loss is computed before the transform takes place. This method works for this model in particular since the transform component of the function does not need to be trained. Thus, standard cross-entropy can be used, since the model is equivariant before the transform is applied.

## C.5 Full Results

All experiments were run in a distributed manner on an internal university cluster, on exclusively CPU nodes. In our final experiments, there were 260 individual runs. A total of 88 days of compute time was used for our final experiments. Across all of our experiments, including failed runs and preliminary experiments, a total of 219 days of compute time was used. Full results for our experiments across each of the datasets are shown in the following sections. The mean is shown along with standard error.

### C.5.1 Bioisostere

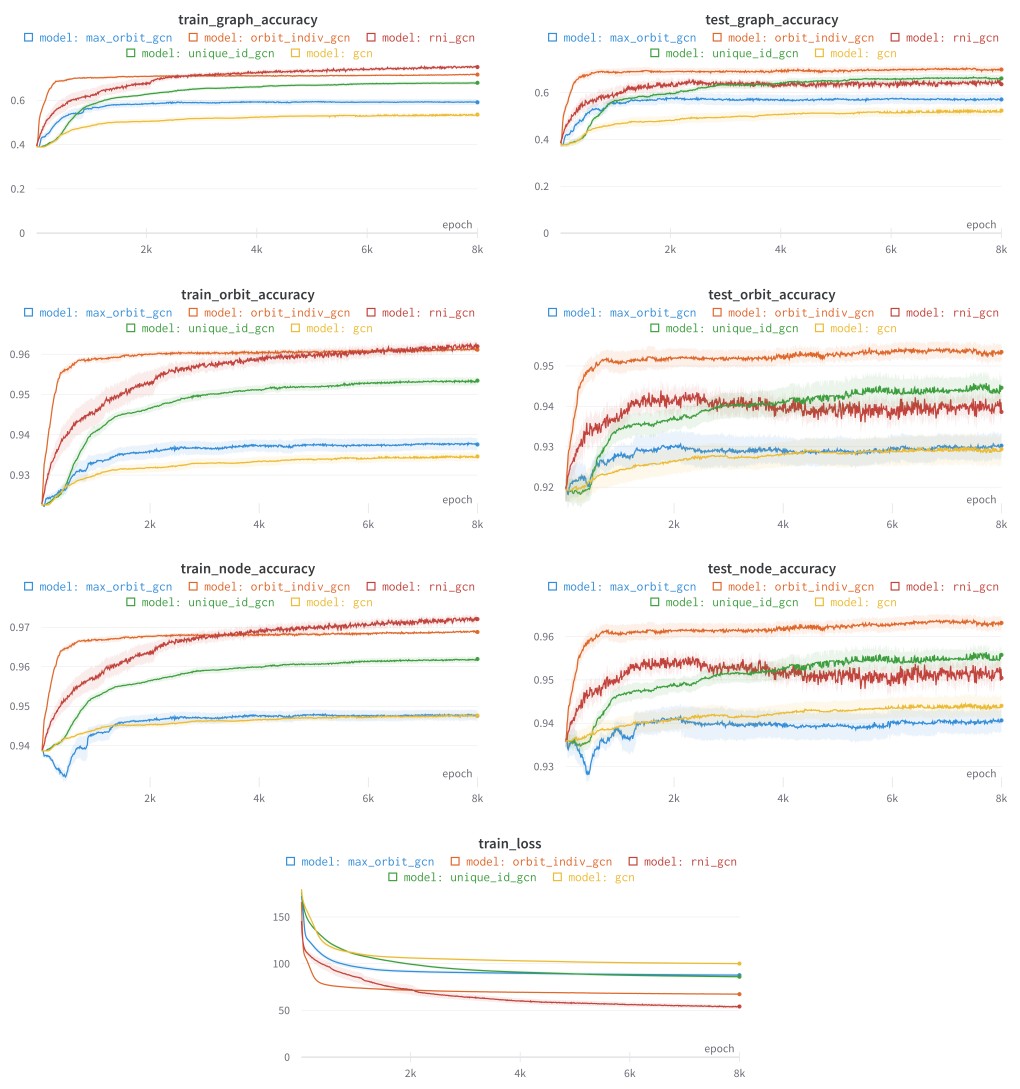

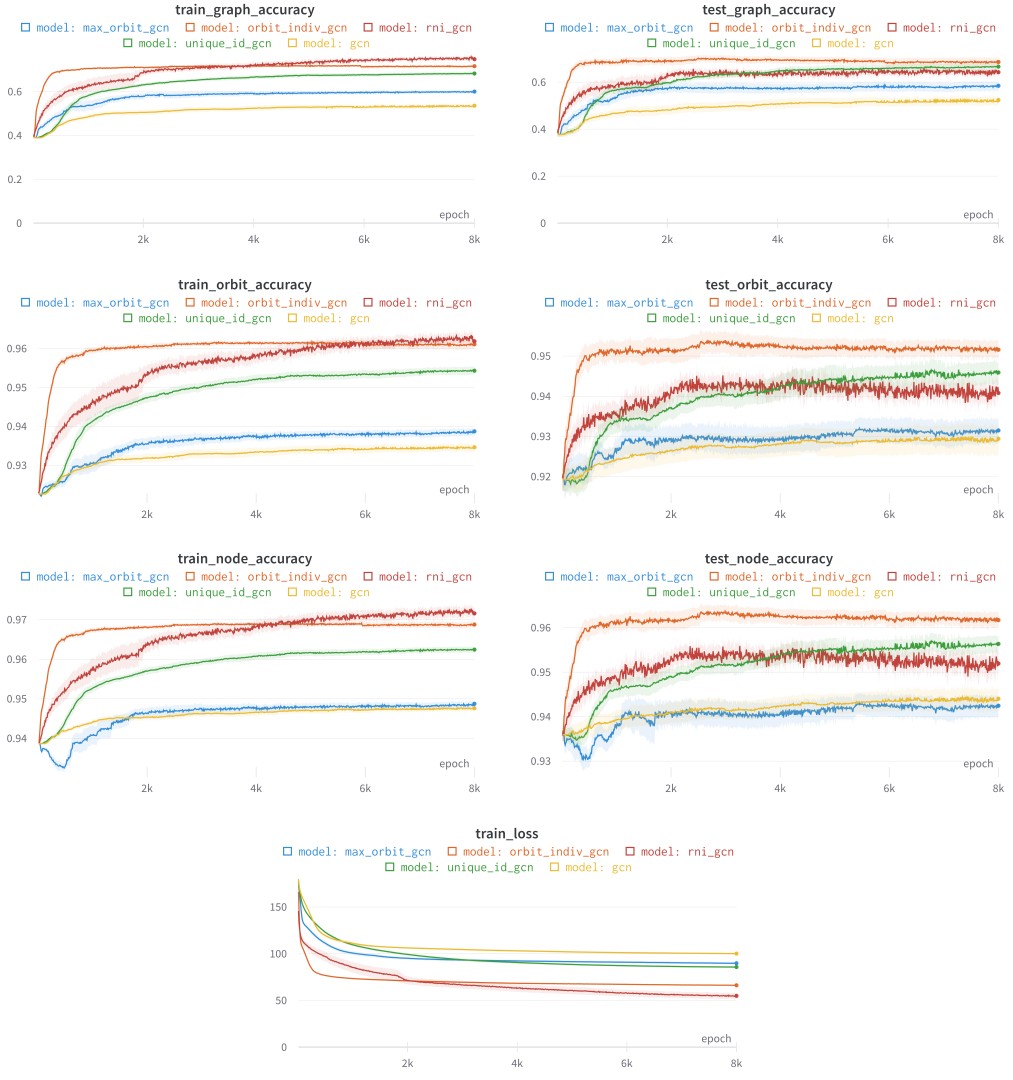

## C.5.2  ALCHEMY-MAX-ORBIT-2

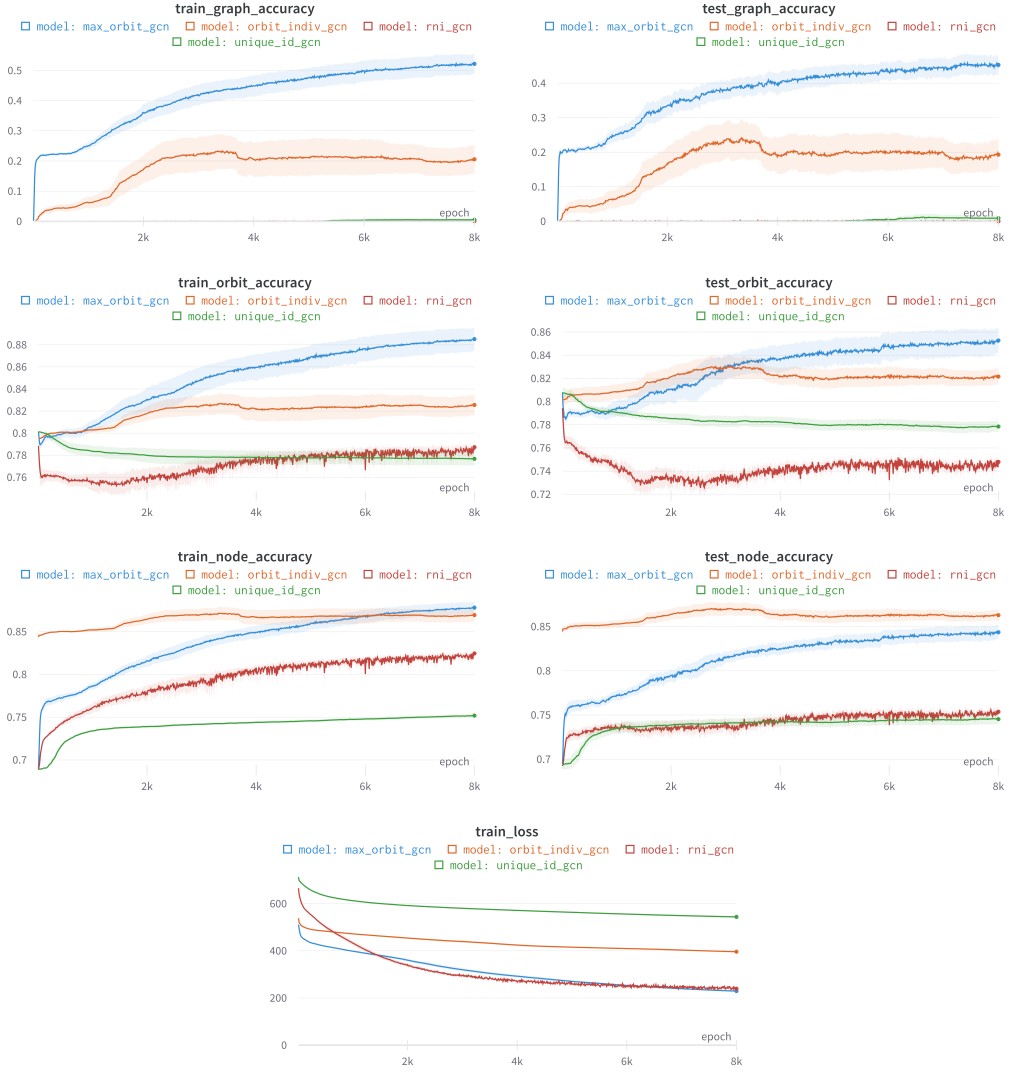

## Orbit-Sorting Cross-Entropy Loss

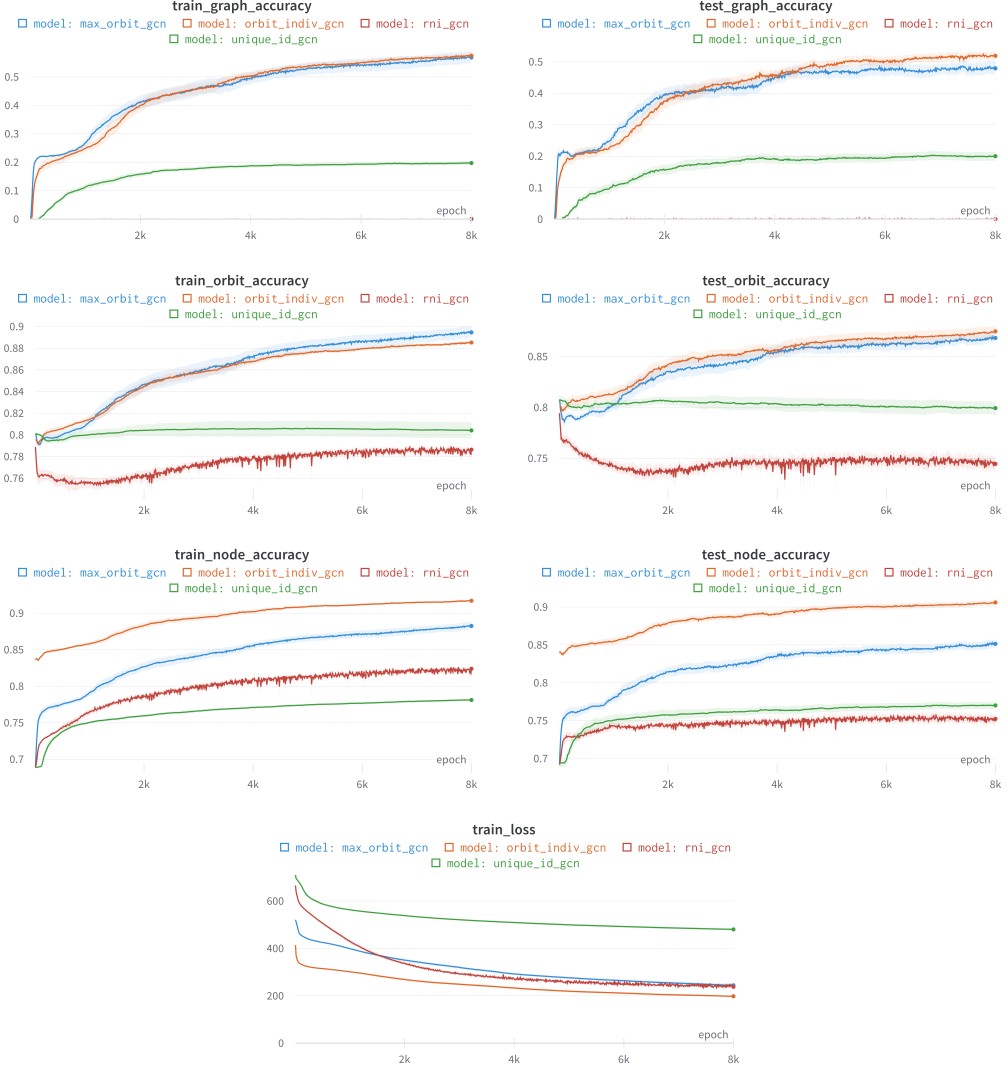

### C.5.3 ALCHEMY-MAX-ORBIT-6

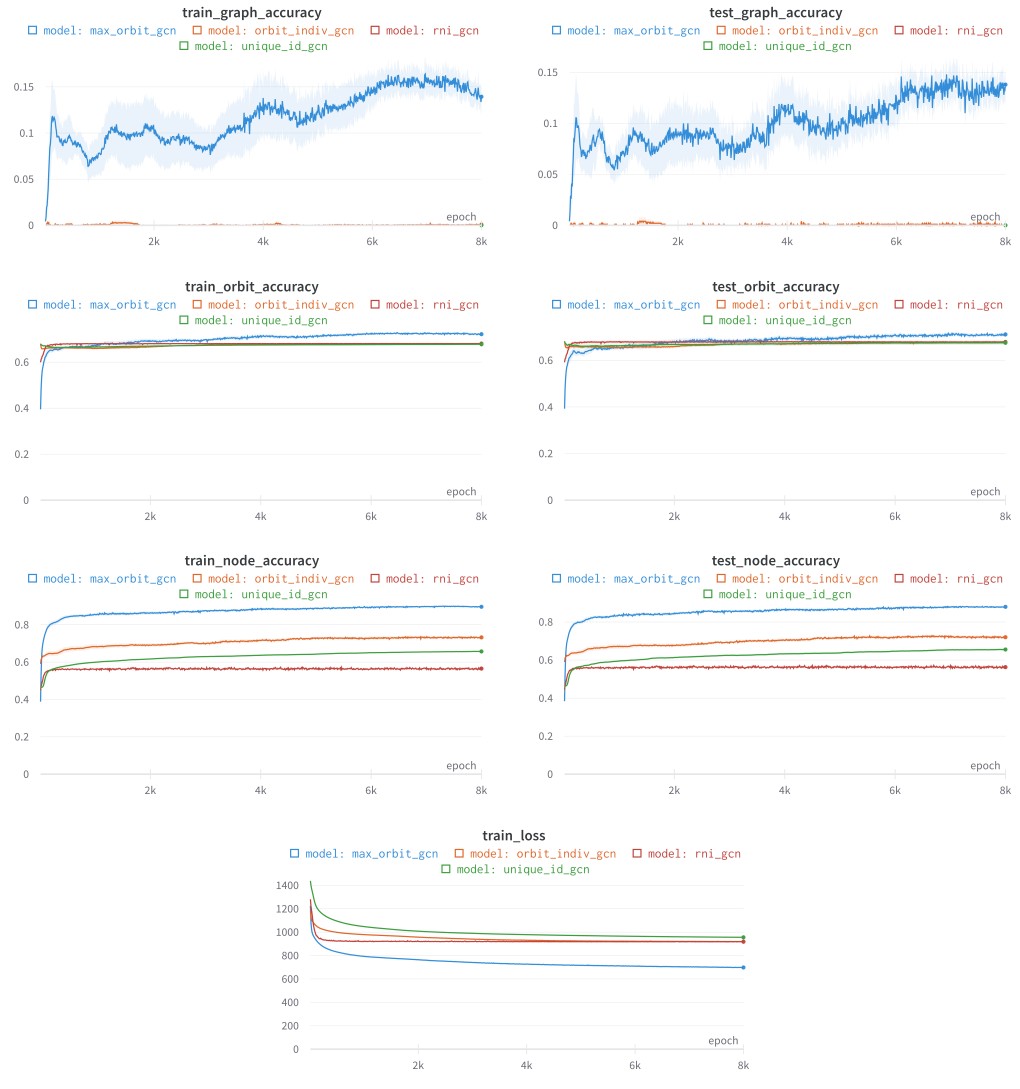

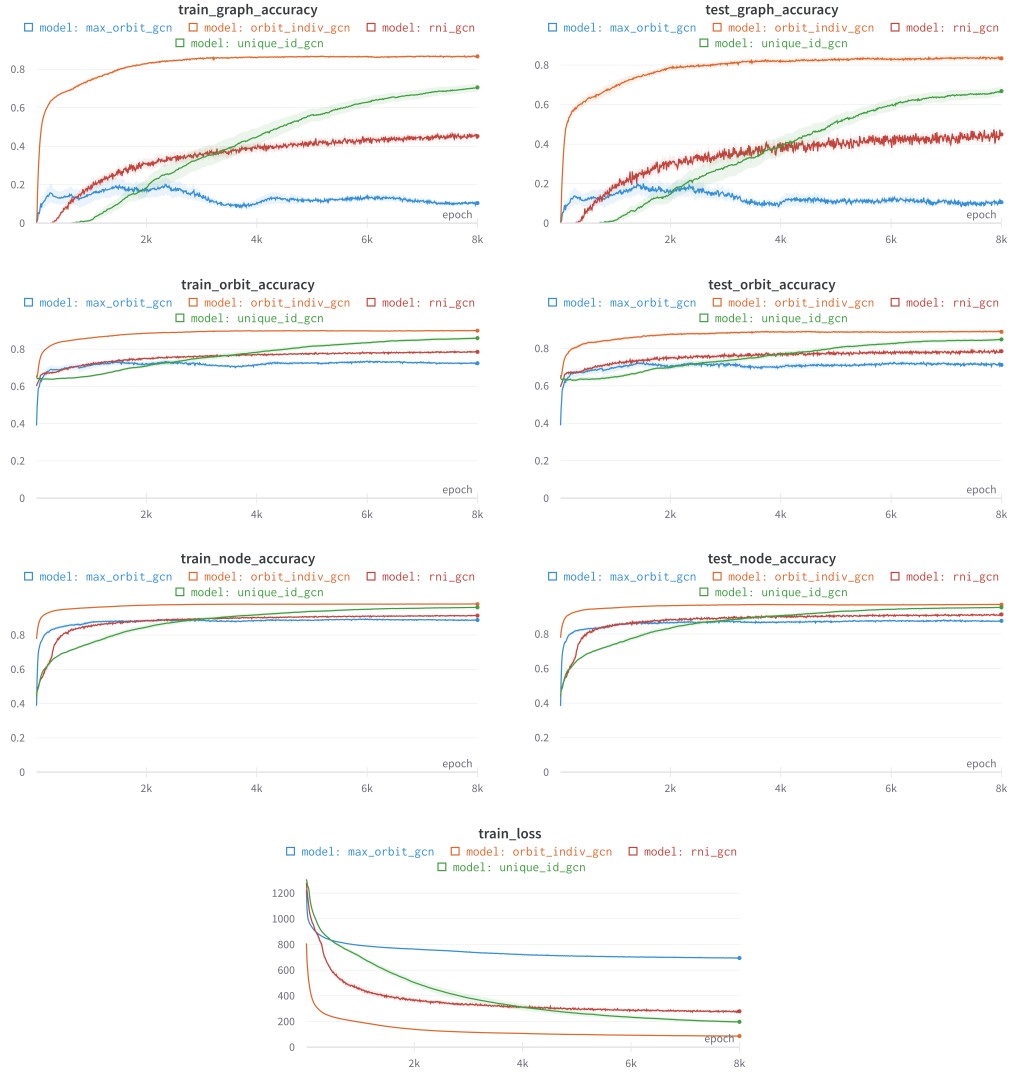

