# OpenReview forum: "Orbit-Equivariant Graph Neural Networks"
_ICLR.cc/2024/Conference — ICLR 2024 poster_

### Official Review · Reviewer_AmbM · 2023-10-31

**Soundness:** 3 good
**Presentation:** 2 fair
**Contribution:** 3 good
**Rating:** 6
**Confidence:** 2

**Summary:**

This paper propose to capture a new kind of equivariance called orbit-equivariance.
The orbit denotes node subsets in graphs.
When a function is orbit equivariance, the label subsets containing labels of nodes in orbit are the same if the input nodes indices are permuted.
For the commonly used GNN, it always preserves permutation equivariance $S_n$, and it is also orbit-equivariance and orbit-equivariance is less restricted.
I suggest to give a better conceptual illustration for max-orbit first, and then provide the motivation to give such max-orbit function to make it more clear.

**Strengths:**

1. The proposed orbit-equivariance is a new symmetry for GNN, and two new datasets are provided with the required symmetry.
2. From the experimental results, the proposed Orbit-Indiv-GCN can achieve much better performance compared to original GCN model on the new datasets.

**Weaknesses:**

1. The figure and illustration of concept is concrete, but it is better if conceptual understanding is provided. The current writing flow is a little hard to follow.
2. The proposed Bioisostere data is obtain by RdKit which is based on semi-emperical methods, and accurate geometry can be obtained through DFT calculation like QM9 and OC20.

**Questions:**

1. Would you mind providing some illustrations about the concept of max-orbit?
2. I am not sure whether the proposed model has some connection to the paper like DeepSet [1]. Would you mind giving some insights?
3. The motivation of developing orbit-equivariance is still confusing to me. Would you mind providing more examples and situations that such symmetries are required?

[1] Zaheer, Manzil, et al. "Deep sets." Advances in neural information processing systems 30 (2017).

---

> ### Author Response · Authors · 2023-11-14
>
> *Question:*
>
> Would you mind providing some illustrations about the concept of max-orbit?
>
> **Response:**
>
> We have adjusted the definition to eliminate potential confusion.
>
> *Question:*
>
> I am not sure whether the proposed model has some connection to the paper like DeepSet. Would you mind giving some insights?
>
> **Response:**
>
> The aim of DeepSets is to provide a learnable function that can operate on sets, such that the functions are permutation invariant. By contrast, our work defines functions on graphs. A model like DeepSets could be used when implementing a GNN, for f_aggr or f_read (see P.3, Section 2, Graph Neural Networks).
>
> *Question:*
>
> The motivation of developing orbit-equivariance is still confusing to me. Would you mind providing more examples and situations that such symmetries are required?
>
> **Response:**
>
> The multi-agent application from Morris et al. (2022) and our drug design application are the main motivating examples we have identified for equivariance being too strict of a property. In general, orbit-equivariance could be beneficial in any application that 1) uses node-labeling functions on graphs to optimize some global property and 2) has nodes that are not already individualized.

---

### Official Review · Reviewer_GKac · 2023-11-06

**Soundness:** 4 excellent
**Presentation:** 3 good
**Contribution:** 3 good
**Rating:** 8
**Confidence:** 4

**Summary:**

Typical GNNs are permutation equivariant, which, however, are unable to allow antisymmetric behaviors of symmetric nodes in the same orbit. Motivated from the example of molecules transformation and agent learning, this paper defines orbit-equivariance, a relaxation of equivariance which allows for such functions whilst retaining important structural inductive biases. Two orbit-equivariant GNNs are proposed namely Orbit-Indiv-GNNs and m-Orbit-Transform-GNNs. Besides the interesting theoretical derivations, the proposed models are evaluated empirically.

**Strengths:**

Strengths:

1. This paper is well written and well-motivated. I enjoy the reading. The example in Figure 1 and the illustrated examples in Section 3 nicely help the understanding of the proposed idea. The propositions and theorems are well organized and clearly demonstrated.

2. The proposed idea of orbit-equivariance is novel and valuable. It allows permutation equivariance between the nodes in different orbits, but breaks the equivariance for the nodes in the same orbit. In this way, while it allows the distinct output of the symmetric nodes in the same orbit, the hierarchy of the graph structure is still represented in an invariant way, which means the multiset of outputs associated with each distinct graph orbit is unchanged. More succinctly, an orbit-equivariant GNN now becomes a set function of orbits other than individual nodes.

3. The design of Orbit-Indiv-GNNs and m-Orbit-Transform-GNNs is meaningful and fulfill the theoretical guarantee derived by Theorem 2.

4. The experiments, although not as impressive as the methodology part, are still sufficient to support the benefit of the proposed orbit-equivariance and the developed models.

**Weaknesses:**

1. The introduction of m-Orbit-Transform-GNN is a bit confusing. I have to read it multiple times to understand the process. It seems there are many freedoms to derive an orbit-equivariant GNN that has max-orbit no more than m. It is unclear why the authors choose the one proposed in the paper. And the experiments on Bioisostere show that the performance of 2-Orbit-Transform-GNN is worse than Orbit-Indiv-GNNs. Does it mean the proposed construction of m-Orbit-Transform-GNN requires further exploration?

2. Another concern is that the processes in both Orbit-Indiv-GNN and m-Orbit-Transform-GNN to make them orbit-equivariant are not differential. I would like to hear form the authors’ thinking about if it is possible to derive a learnable equivariance-to-orbit-equivariant layer other the ad-hoc transformations proposed by the authors. Will the learnable ones further improve the performance?

3. The last concern is about the efficiency of Orbit-1-WL that is applied to determine the orbits, particularly for large-scale graphs in practice. The authors should discuss this point if necessary.

**Questions:**

No more question but the concerns in the weakness part.

---

> ### Author Response · Authors · 2023-11-14
>
> *Question:*
>
> The experiments on Bioisostere show that the performance of 2-Orbit-Transform-GNN is worse than Orbit-Indiv-GNNs. Does it mean the proposed construction of m-Orbit-Transform-GNN requires further exploration?
>
> **Response:**
>
> Whilst the results in P.8, Table 1 and P.9, Figure 7 show that 2-Orbit-Transform-GCN is at least competitive with Orbit-Indiv-GCN on Alchemy-Max-Orbit-2, the results on Bioisostere do suggest that it could be further investigated in future work.
>
> *Question:*
>
> Another concern is that the processes in both Orbit-Indiv-GNN and m-Orbit-Transform-GNN to make them orbit-equivariant are not differential.
>
> **Response:**
>
> The transform operation in m-Orbit-Transform-GNN does not need to be differentiable, since it is not trained; this is noted in P.7, Section 5, Loss. The output of an Orbit-Indiv-GNN is differentiable with respect to its parameters, so it can be trained without issues.
>
> *Question:*
>
> I would like to hear from the authors’ thinking about if it is possible to derive a learnable equivariance-to-orbit-equivariant layer other than the ad-hoc transformations proposed by the authors. Will the learnable ones further improve the performance?
>
> **Response:**
>
> Orbit-Indiv-GNN is a simple example of a learnable equivariant-to-orbit-equivariant layer. Other types of learnable layers could exist, but this is left for future work.
>
> *Question:*
>
> The last concern is about the efficiency of Orbit-1-WL that is applied to determine the orbits, particularly for large-scale graphs in practice.
>
> **Response:**
>
> In P.5, Section 4, we note that Appendix C.1 has an analysis of the computational complexity of the models. There, it is pointed out that 1-WL is worst-case linear in the number of nodes in the graph. Thus, we anticipate that Orbit-1-WL will scale well to large graphs.

---

### Official Review · Reviewer_y9Hz · 2023-11-06

**Soundness:** 3 good
**Presentation:** 2 fair
**Contribution:** 3 good
**Rating:** 6
**Confidence:** 2

**Summary:**

The paper first introduces the orbit-equivariance of a function that takes a graph (with an ordered node set) as the invariance of the multiset of values on each orbit of the graph isomorphism under node permutations. Also, $\texttt{max-orbit}$ is defined as the maximum number of unique values on each orbit, and it is shown that the usual equivariance is equivalent to orbit-equivariance and $\texttt{max-orbit}=1$. Two types of existing GNNs are shown to be orbit-equivariant either deterministically or stochastically. Also, Orbit-Indiv-GNN and $m$-Orbit-Transform-GNN are proposed, which are orbit-equivariant, along with orbit-sorting cross-entry loss as a loss function for orbit-equivariant functions to train the models. The proposed models are applied to real or synthetic datasets that require orbit equivariance to verify their practical prediction performances.

**Strengths:**

- The significance of introducing the concept of orbit-equivariant functions is appropriately demonstrated by using an example from drug discovery. The mathematical definition of orbit-equivariance is appropriate because it adequately explains the drug discovery example.
- By introducing the concept of max-orbit, the relationship between orbit-equivariance and equivariance is clearly shown (Proposition 3).
- This paper shows the existence of GNNs that are orbit-equivariant by concretely constructing Orbit-Indiv-GNN and m-Orbit-Transform-GNN.
- Numerical experiments show that the proposed models improve accuracy in tasks that require learning orbit-equivariant functions.
- The hyperparameters and datasets are described in detail, and the code is provided, making the experiments reproducible.

**Weaknesses:**

- The description of Orbit-Transform-GNN (Figure 5) has room for improvement (see Questions).
- I have a question about whether the learning method of $m$-Orbit-Transform-GNN is appropriate (see Questions).
- The proposed $m$-Orbit-Transform, although constructive, is constructed using operations that are difficult to explain intuitively, such as rewriting output values. In addition, numerical experiments have shown that $m$-Orbit-Transform has yet to achieve good accuracy.

**Questions:**

* P.2, Section 2: It is appropriate that the input to an orbit-equivariant function (or, more generally, a node-labeling function) is a graph with an ordered set of nodes. Node sets of graphs in this paper are ordered as $V=\\\{1,\ldots, N\\\}$. Also, examples of node-labeling functions are defined using this ordering. If the node-labeling function takes the graph structure only, we need to show that the function is well-defined regardless of the node ordering. If it is well-defined, the function is automatically equivariant.
* P.6, Figure 5: Since $m=o+1(=3)$, it is difficult to understand whether the triples represent $m$ or $o+1$. It would be better to use a different number for both (e.g., $m=5$). In Figure 5, the last element of the partition is 1-index, whereas 0-index is used in the text, which is confusing and should be unified.
* P.6, Theorem 2: Regarding the claim that a function is not orbit-equivariant, does it mean the function is *not necessarily* orbit-equivariant? For example, Unique-ID-GNNs happen to be orbit-equivariant depending on the choice of the underlying GNN (e.g., degenerated GNN that always returns 0.)
* P.7, Section 5: I have a question about whether it is appropriate to apply the cross-entropy loss to the output before transforming in the training of $m$-Orbit-Transform-GNN. This operation implies that $m$-Orbit-Transform-GNN is trained to output the correct labels before the transformation. If I understand correctly, this is different from what is intended.
* P.8, Section 5: GCNs [...] would have achieved an accuracy of 0 on the Alchemy-Max-Orbit, [...]: Does this sentence mean that GCNs have not been tested on the Alchemy-Max-Orbit?
* P.8, Table 1: The graph accuracy of RNI-GCN is 0 for all ten trials in Alchemy-Max-Orbit-2. Is this result due to the nature of RNI-GCN? Or does insufficient training cause it, and RNI-GCN could have non-zero accuracy when appropriately trained?

【Minor Comments】
* P.1, Section 1: Graph Neural Networks (GNNs) -> GNNs: The abbreviation for GNNs appears at the beginning of this section. So, there is no need to repeat the full phrase.
* P.2, Section 2: aka -> a.k.a. or also known as (I suggest using the full phrase)
* P.3, Section 2: $f^{\theta'\_{m}}\_{\mathrm{aggr}}$ and $f^{\theta''\_{m}}\_{\mathrm{read}}$ are imposed to be permutation invariant, but both of them are automatically equivariant when receiving a multi-set Isn't it automatically a permutation invariant when both of them receive a multi-set?
* P.4, Section 3: $\\\{0\\\}^{|G|}$ -> $0^{|G|}$
* P.4, Section 3: the number of unique values in $\\\{\\\{ f(G)_v \mid v\in r \\\}\\\}$: wouldn't it be easier to write $|\\\{f(G)_v \mid v\in r\\\}|$ in a mathematical way?
* P.5, Section 4: What does *Indiv* in Orbit-Indiv-GNN stand for? Does it mean individualization?
* P.7, Section 5: $|T|$ is not used.

**Details Of Ethics Concerns:**

N.A.

---

> ### Author Response · Authors · 2023-11-14
>
> *Question:*
>
> If the node-labeling function takes the graph structure only, we need to show that the function is well-defined regardless of the node ordering. If it is well-defined, the function is automatically equivariant.
>
> **Response:**
>
> We use V = {1, …, N} in our paper for ease of notation only: it is not required, yet it is a very common notation. Giving a set of nodes is needed to properly define a graph: thus, functions on graphs need to take this into account. The first example function shown on P.4 is a well-defined node-labeling function on a graph that is not equivariant.
>
> *Question:*
>
> P.6, Figure 5: Since m = o + 1 (=3), it is difficult to understand whether the triples represent m or o +  1. It would be better to use a different number for both (e.g., m = 5). In Figure 5, the last element of the partition is 1-index, whereas 0-index is used in the text, which is confusing and should be unified.
>
> **Response:**
>
> We have added an extra annotation to show which triple corresponds to o + 1 = 3. Hopefully this eliminates the potential confusion. We have also fixed the inconsistency between the figure and text, in the 0-indexing vs 1-indexing.
>
> *Question:*
>
> P.6, Theorem 2: Regarding the claim that a function is not orbit-equivariant, does it mean the function is not necessarily orbit-equivariant?
>
> **Response:**
>
> Yes. By the statement “a class of models X is not orbit-equivariant”, we mean that there exists a function f from the class X that is not orbit-equivariant.
>
> *Question:*
>
> P.7, Section 5: I have a question about whether it is appropriate to apply the cross-entropy loss to the output before transforming in the training of m-Orbit-Transform-GNN.
>
> **Response:**
>
> Yes, we do intend to apply the loss before the transformation takes place. We have revised P.7, Section 5, Loss to make this clearer.
>
> *Question:*
>
> P.8, Section 5: GCNs [...] would have achieved an accuracy of 0 on the Alchemy-Max-Orbit, [...]: Does this sentence mean that GCNs have not been tested on the Alchemy-Max-Orbit?
>
> **Response:**
>
> Yes, GCNs are not expressive enough to solve any of the examples in the Alchemy-Max-Orbit datasets.
>
> *Question:*
>
> P.8, Table 1: The graph accuracy of RNI-GCN is 0 for all ten trials in Alchemy-Max-Orbit-2. Is this result due to the nature of RNI-GCN or rather how it was trained?
>
> **Response:**
>
> The full results in P.37, Section C.5.2, Orbit-Sorting-Cross-Entropy Loss show that the performance of RNI-GCNs plateaus by 8k epochs (under all metrics), so it appears unlikely that further training would improve the model. However, there is nothing inherent to the nature of the model that strictly prevents it from achieving a non-zero graph accuracy.
>
> *Question:*
>
> P.5, Section 4: What does Indiv in Orbit-Indiv-GNN stand for? Does it mean individualization?
>
> **Response:**
>
> Yes: it refers to individualization within each orbit.
>
> *Question:*
>
> P.7, Section 5: |T| is not used.
>
> **Response:**
>
> We have rephrased this sentence to eliminate the confusion.
>
> *Other Minor Comments*
>
> **Response:**
>
> Thank you for paying such close attention to our paper to have made these comments. We have addressed them in our uploaded revision.

---

> > ### Comment · Reviewer_y9Hz · 2023-11-22
> > **Response to authors' comments**
> >
> > I thank the authors for answering my review comments. Here, I respond to the authors' comments one by one.
> >
> > > We use V = {1, …, N} in our paper for ease of notation only: it is not required, yet it is a very common notation. Giving a set of nodes is needed to properly define a graph: thus, functions on graphs need to take this into account. The first example function shown on P.4 is a well-defined node-labeling function on a graph that is not equivariant.
> >
> > I think the function $f$ use node ordering. Let $G_4$ be the graph created from $G_1$ by swapping node 2 and 3:
> >
> > ```
> >   3
> >  /
> > 1
> >  \
> >   2
> > ```
> >
> > By the definition of $f$, $f(G_4) = 0$. However, we cannot distinguish $G_1$ and $G_4$ solely from the graph structure (i.e., without the specific order of the node set.)
> >
> > --------------
> >
> >
> > > We have added an extra annotation to show which triple corresponds to o + 1 = 3. Hopefully this eliminates the potential confusion. We have also fixed the inconsistency between the figure and text, in the 0-indexing vs 1-indexing.
> >
> > OK
> >
> > --------------
> >
> > > Yes. By the statement “a class of models X is not orbit-equivariant”, we mean that there exists a function f from the class X that is not orbit-equivariant.
> >
> > OK. It is better to clarify it because the statement can mistekenly interpret that any function in the class of models X is *always* non-orbit-equivariant.
> >
> > --------------
> >
> > > Yes, we do intend to apply the loss before the transformation takes place. We have revised P.7, Section 5, Loss to make this clearer.
> >
> > OK. I understand that for $m$-OrbitTransform-GNN, the output before transformation is compared with the inverse-transform of the targets. I want to confirm the motivation for this loss design because we can choose to compute the loss after the transformation using orbit-sorting cross-entropy loss.
> >
> > --------------
> >
> > > Yes, GCNs are not expressive enough to solve any of the examples in the Alchemy-Max-Orbit datasets.
> >
> > OK. I agree that GCN cannot solve Alchemy-Max-Orbit-2 and Alchemy-Max-Orbit-6 well (strictly speaking, the node accuracy can be a positive value. But I understand that the graph accuracy and orbit accuracy can be 0)
> >
> > --------------
> >
> > > The full results in P.37, Section C.5.2, Orbit-Sorting-Cross-Entropy Loss show that the performance of RNI-GCNs plateaus by 8k epochs (under all metrics), so it appears unlikely that further training would improve the model. However, there is nothing inherent to the nature of the model that strictly prevents it from achieving a non-zero graph accuracy.
> >
> > OK
> >
> > --------------
> >
> > > Yes: it refers to individualization within each orbit.
> >
> > OK, I would suggest clarifying what Indiv stands for.
> >
> > --------------
> >
> > > We have rephrased this sentence to eliminate the confusion.
> >
> > OK

---

> > > ### Author Response · Authors · 2023-11-22
> > > **Response to follow-up**
> > >
> > > We would like to thank the reviewer for their further engagement. Here, we respond to the suggestions and queries raised.
> > >
> > > > I think the function $f$ use node ordering. Let $G_4$ be the graph created from $G_1$ by swapping node 2 and 3. By the definition of $f$, $f(G_4) = 0$. However, we cannot distinguish $G_1$ and $G_4$ solely from the graph structure (i.e., without the specific order of the node set.)
> > >
> > > Thank you for the example. The graph you have provided ($G_4$) is equal to $G_1$ (not just isomorphic), since it has the same sets of vertices and edges, and thus $f(G_4) = f(G_1) = (0, 1, 0)$. This is different to $G_2$, which is isomorphic to $G_1$ but not equal. Hopefully this helps illustrate our definitions of graph functions and graph equality.
> > >
> > > > OK. It is better to clarify it because the statement can mistakenly interpret that any function in the class of models X is always non-orbit-equivariant.
> > >
> > > Thank you for raising this. We have edited Theorem 2 in our revision to remove this ambiguity.
> > >
> > > > OK. I understand that for $m$-OrbitTransform-GNN, the output before transformation is compared with the inverse-transform of the targets. I want to confirm the motivation for this loss design because we can choose to compute the loss after the transformation using orbit-sorting cross-entropy loss.
> > >
> > > It is true that we could choose to instead use orbit-sorting cross-entropy loss (as we do for the other models), but since using a novel loss function compared to the provenly effective cross-entropy loss is less desirable, $m$-OrbitTransform-GNNs provide a way to make use of the established cross-entropy loss function.
> > >
> > > > OK. I agree that GCN cannot solve Alchemy-Max-Orbit-2 and Alchemy-Max-Orbit-6 well (strictly speaking, the node accuracy can be a positive value. But I understand that the graph accuracy and orbit accuracy can be 0)
> > >
> > > Great point. In our revision on P.7, Section 5, Methodology, we have clarified that "GCNs are equivariant and thus cannot achieve a better *graph* accuracy than 0 on the Alchemy-Max-Orbit datasets".

---

> > > > ### Comment · Reviewer_y9Hz · 2023-11-23
> > > > **About issues related to node orderings**
> > > >
> > > > I thank the authors for the detailed explanation.
> > > >
> > > > Regarding the issue of node ordering and identification of graphs, I think that the authors and I are equivalent.
> > > >
> > > > Let us first ignore the node ordering and define a graph $G=(V, E)$ by $V=\\{v_1, v_2, v_3\\}$ and $E=\\{\\{v_1, v_2\\}, \\{v_1, v_3\\}\\}$.
> > > > We define three node orderings (more specifically, mappings of node sets to the set $\\{1, \ldots, |V|\\}$), namely, $a_1$, $a_2$, and $a_4$ by
> > > >
> > > >
> > > > $$
> > > > \begin{align}
> > > > a_1\colon v_1\mapsto 1, v_2\mapsto 2, v_3\mapsto 3,\\\\
> > > > a_2\colon v_1\mapsto 2, v_2\mapsto 1, v_3\mapsto 3,\\\\
> > > > a_4\colon v_1\mapsto 1, v_2\mapsto 3, v_3\mapsto 2.
> > > > \end{align}
> > > > $$
> > > >
> > > > When we apply $G$ to $a_i$, the resulting graphs are $G_i$ ($i=1, 2, 4$). Therefore, if we ignore the node ordering, $G_i$'s are the identical (namely, $G$)
> > > > On the other hand, $G_1$ and $G_4$ are identical in the sense that both graphs have the node set $(1, 2, 3)$ and the edge set $\\{(1, 2), (1, 3)\\}$, while $G_1$ ($=G_4$) and $G_2$ are different.
> > > > Therefore, I think it is impossible to distinguish the identity of two graphs and the isomorphism of two graphs without the order of the node set and that assuming $V = \\{1, \ldots, |V|\\}$ implicitly imposes an order on the node-set.
> > > >
> > > > I want to note that the mathematical statements in this paper are correct and that the paper does not have to change its main claims.

---

### Official Review · Reviewer_nEVg · 2023-11-09

**Soundness:** 3 good
**Presentation:** 4 excellent
**Contribution:** 3 good
**Rating:** 8
**Confidence:** 3

**Summary:**

The authors propose an overarching framework for the hierarchy of node-labelling functions via the lens of orbit-equivariances. They do this by allowing an equivalence class of nodes in a graph to be mapped to a multiset of outputs. The hierarchy spans from permutation equivariant functions for graphs (two nodes belong to the same equivalence class obtain the same representation) to the case where all nodes get unique representations (aking to positional embeddings). The authors  introduce max-orbit, which allows for further control of
orbit-equivariant functions - and establish a theoretical connection between 1-Weisfeiler Leman isomorphism test to identification of orbits and thereby categorizing the expressive power of different GNN architecture. Subsequently they study 4 different GNN architectures in terms of theoretical expressiveness. In the experimental front, the authors propose two new datasets to demonstrate the success of using orbit-equivariant GNNs.

**Strengths:**

1. The paper proposes a hierarchy which brings under one umbrella different GNN architectures.
2. The paper provides a study of theoretical expressiveness (akin to the WL hierarchy)
3. The paper shows empirical evidence on simple new datasets (proposed by the authors) to validate their claims
4. The paper is very well written and easy to comprehend

**Weaknesses:**

**Minor Weakness**
1. Misses relevant theoretical works which theoretically unifies positional embedding GNNs - to equivariant GNNs, and node labelling works to develop more powerful GNNs [1][2][3]
2. In figure 1, please make it explicit that the 3 Fluorine atoms in the molecule belong to the same equivalence class only if the position coordinates of the atoms - are not used as part of the node features
3. As the authors list in the limitations section - there is a lack of strong experimental results present in many real word molecular datasets, etc.
4. Novelty is definitely present - but not something completely unexpected and draws and builds upon existing literature


**References**

[1] Srinivasan and Ribeiro, ICLR 2020, On the Equivalence between Positional Node Embeddings and Structural Graph Representations.

[2] Li, et al. Neurips 2020, Distance Encoding: Design Provably More Powerful Neural Networks for Graph Representation Learning

[3] Zhang, et al. Neurips 2021, Labeling Trick: A Theory of Using Graph Neural Networks for Multi-Node Representation Learning

**Questions:**

Please address the minor weaknesses in the prior section.

Additionally, the term orbit-equivariance appears to be a slightly misleading name for the framework - given it is not about learning representations for the elements in an orbit  which are equivariant to something like a permutation action on the orbit itself. But it is rather assigning a multiset as output labels to an orbit. Unfortunately, I do not have an apt name as well - but would suggest the authors ponder about this a bit more.

---

> ### Author Response · Authors · 2023-11-14
>
> *Question:*
>
> The paper is missing relevant theoretical works [1][2][3].
>
> **Response:**
>
> We thank the reviewer for referring us to these useful related works. We have included references and discussion about them in the introduction and conclusion.
>
> *Question:*
>
> In figure 1, please make it explicit that the 3 Fluorine atoms are only in the same orbit if they are not labeled with positional information.
>
> **Response:**
>
> We have clarified this in the caption of Figure 1.

---

> > ### Comment · Reviewer_nEVg · 2023-11-14
> > **Acknowledge the rebuttal**
> >
> > Dear Authors,
> >
> > Thank you for the rebuttal. Having gone through all the other reviews as well as your responses, I will stick to my scores of accept (8).

---

### Author Response · Authors · 2023-11-14
**Proposed Changes**

We would like to thank all of the reviewers of our paper for such thorough engagement and insightful comments.

Based on the feedback received from all reviewers so far, we propose to make the following changes in the revised version of our paper:
- Additional discussion of related works in the introduction and conclusion.
- Clarify in Figure 1 that the 3 Fluorine atoms are only in the same orbit if they are not labeled with positional information.
- P.1, Section 1: remove redundant definitions of GNN acronym
- P.2, Section 2: use full phrase instead of “aka”
- P.4, Section 3, the number of unique values in the multiset when defining max-orbit: included cardinality of the set in the definition
- In Figure 5, add an extra annotation to show which triple corresponds to o + 1 = 3. Also fix the inconsistency between the figure and text, in the 0-indexing vs 1-indexing.
- Revise P.7, Section 5, Loss to make the training of m-Orbit-Transform-GNN clearer.
- P.7, Section 5: rephrase the definition of T to eliminate confusion
- Cut some minor content from the introduction to stay within the space limits

These changes have been applied in the uploaded revised version of our paper, and temporarily highlighted in red to make the reviewing process easier.

---

### Meta-Review · Area_Chair_hg8Z · 2023-12-15

**Metareview:**

There was an active discussion phase with the reviewers and after the rebuttal also with the AC. All agreed in the end that the paper is interesting, novel, and has strong results.

**Justification For Why Not Higher Score:**

Several (minor) weakness were pointed out by the reviewers/

**Justification For Why Not Lower Score:**

All reviewers agreed that the paper should be accepted.

---

### Decision · Program_Chairs · 2024-01-16

Accept (poster)